# EUV-induced hydrogen desorption as a step towards large-scale silicon quantum device patterning

Procopios Constantinou [1,2,3] ✉, Taylor J. Z. Stock [1,4], Li-Ting Tseng[3], Dimitrios Kazazis [3], Matthias Muntwiler [3], Carlos A. F. Vaz [3], Yasin Ekinci[3], Gabriel Aeppli [3,5,6,7], Neil J. Curson [1,4] & Steven R. Schofield [1,2] ✉

Atomically precise hydrogen desorption lithography using scanning tunnelling microscopy (STM) has enabled the development of single-atom, quantum-electronic devices on a laboratory scale. Scaling up this technology to mass-produce these devices requires bridging the gap between the precision of STM and the processes used in next-generation semiconductor manufacturing. Here, we demonstrate the ability to remove hydrogen from a monohydride Si(001):H surface using extreme ultraviolet (EUV) light. We quantify the desorption characteristics using various techniques, including STM, X-ray photoelectron spectroscopy (XPS), and photoemission electron microscopy (XPEEM). Our results show that desorption is induced by secondary electrons from valence band excitations, consistent with an exactly solvable non-linear differential equation and compatible with the current 13.5 nm (~92 eV) EUV standard for photolithography; the data imply useful exposure times of order minutes for the 300 W sources characteristic of EUV infrastructure. This is an important step towards the EUV patterning of silicon surfaces without traditional resists, by offering the possibility for parallel processing in the fabrication of classical and quantum devices through deterministic doping.

The future of electronic devices is expected to rely on the principles of quantum logic and reduced-dimension physics, which will enable atomic-scale interactions of spin or charge to support information processing. Breakthroughs in the precise atomic-scale patterning of phosphorous[1], arsenic[2] and boron[3,4] in silicon are making it increasingly possible to create such proposed architectures for silicon-based quantum computation[5,6]. Currently, scanning tunnelling microscopy (STM) based hydrogen desorption lithography[7–9] is used to fabricate laboratory-scale, quantum-electronic devices in silicon consisting of small numbers of dopant atoms[10–13]. However, despite its success, the slow, serial patterning of the STM remains a significant bottleneck for the upscaling to commercial devices that require well-aligned, micrometre-scale dopant arrays and gates for controlling the mutual interactions of qubits[14].

In addition to high-precision hydrogen desorption using STM, it is known that hydrogen desorption can also be performed using the electron beam of a scanning electron microscope (SEM) with a lower resolution (micrometre rather than ångstroms)[15–17]. SEM has the benefit of being able to cover larger writing areas ($\mu m^2$ to $mm^2$) at a much faster speed than STM, making it effective for fabricating electrical contacts and interconnects. However, EUV photons (Fig. 1a) have several technical advantages over electron beams: (i) Diffraction optics

[1]London Centre for Nanotechnology, University College London, WC1H 0AH London, UK. [2]Department of Physics and Astronomy, University College London, WC1E 6BT London, UK. [3]Paul Scherrer Institute, 5232 Villigen PSI, Switzerland. [4]Department of Electronic and Electrical Engineering, University College London, London WC1E 7JE, UK. [5]Institute of Physics, Ecole Polytechnique Fédérale de Lausanne (EPFL), 1015 Lausanne, Switzerland. [6]Department of Physics, ETH Zürich, 8093 Zürich, Switzerland. [7]Quantum Center, Eidgenössische Technische Hochschule Zurich (ETHZ), 8093 Zurich, Switzerland. ✉e-mail: procopios.constantinou@psi.ch; s.schofield@ucl.ac.uk

allows for interference-based parallel lithography[18,19], which can achieve high precision and resolution over large areas; (ii) Since photons carry no charge, they interact less with residual contaminants in vacuum, resulting in cleaner devices and are insensitive to stray electric or magnetic fields in the fabrication chamber; (iii) Given the recent move towards EUV lithography by industry to target 5 nm nodes, developing a photon-based method, which has been the workhorse of high-volume semiconductor manufacturing, that is compatible with atomic-scale STM-based lithography has obvious technological and economic advantages. Alternatively, photothermal hydrogen desorption has recently been demonstrated with UV photons as a viable method of fast, large-scale patterning[20]. However, the pattern edge roughness is of micrometre order and precise windows of operation are required to avoid the roughening of the surface during the patterning.

Direct photon desorption of hydrogen on silicon[21–23] involves a σ→σ* transition of the Si-H bond[24,25] that occurs with maximum efficiency at low photon energies comparable to the bonding-antibonding energy separation (~7.9 eV). Hydrogen patterning with such low-energy photons presents unworkable restrictions on the patterning resolution due to the long wavelengths involved (~157 nm). For any practical application, we must turn to X-ray-induced hydrogen desorption. To date, investigations of photon-induced hydrogen

desorption from Si(001):H have had limited success using 1840 eV[26], 100–650 eV[27], 110–112 eV[28] and 20–30 eV photon irradiations[28,29]. On most occasions, some level of hydrogen desorption was reported, with Auger stimulated desorption being the most common mechanism attributed for rupturing the Si-H bonds: for example, the KLL-2LVV Auger electrons at 1840 eV, originating from Si 1s excitations[26], and the LVV Auger electrons at 110–112 eV from Si 2p excitations[28]. Valence band excitations using 20–30 eV photons have also been observed to break the Si-H bond, but, in this case, it is not known whether the desorption is mediated directly by the incident photons, a VVV core-hole Auger relaxation process, or ejected secondary electrons[28,29]. Only a single study of photon-induced hydrogen desorption using synchrotron radiation on a monohydride Si(001):H surface has been reported[27]. In this study, monochromatic irradiation between 100 to 650 eV was found to not remove hydrogen; however, irradiation with non-monochromatic photons led to indirect evidence of hydrogen desorption, which was attributed to either the ~7.9 eV photons present in the lower end of the non-monochromatic spectral distribution of the bending magnet, or to secondary electrons.

The present paper reports a comprehensive study of EUV photon-induced hydrogen desorption from the monohydride Si(001):H surface. We characterise the desorption process by combining in situ STM, X-ray photoelectron spectroscopy (XPS) and photoemission

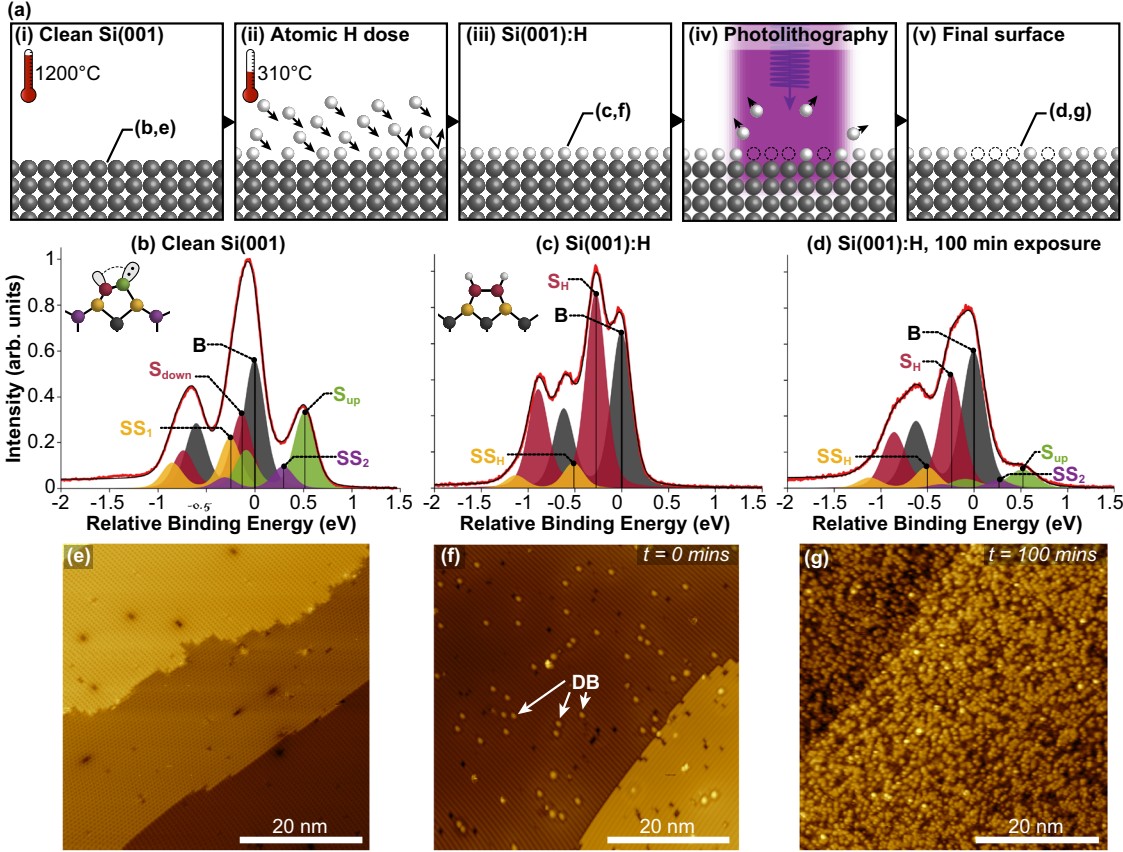

**Fig. 1 | Photon-based hydrogen desorption lithography characterised with combined XPS and STM. a** Process flow of photon-based hydrogen desorption lithography. In steps (i), (iii) and (v), in situ XPS and STM experiments were performed, whose data are presented in (**b, e**), (**c, f**) and (**d, g**), respectively.
**b** Photoelectron spectrum of the clean Si(001) surface. The five fitted components $S_{up}$, $S_{down}$, $SS_1$, $SS_2$ (520, −150, −240 and 240 meV) and $B$ are all individually identified, and colour-coded to the surface reconstruction shown to the left. The uppermost dimers are buckled due to the charge transfer of the π-orbital.
**c** Photoelectron spectrum of the hydrogen terminated Si(001):H surface. The three fitted components $S_H$, $SS_H$ and $B$ are all individually identified (with an energy shift

of −250 and −450 meV, respectively) and colour-coded to the surface reconstruction to the left. **d** Photoelectron spectrum of the Si(001):H surface after 100 min of non-monochromatic irradiation, revealing the emergence of dangling bonds (DBs). All XPS data are for a photon energy $h\nu = 140$ eV at $\theta = 60°$ and the binding energies are measured relative to the bulk component, $B$. The solid red and black lines show the raw and fitted data, respectively, prior to background subtraction, whose maximum peak height is scaled to one. Each fit component is plotted after background subtraction and each energy shift has an uncertainty of 50 meV.
**e–g** Corresponding STM data (−2.5 V, 50 pA) taken on the same region as the XPS for each surface.

electron microscopy (PEEM). We establish that for EUV irradiation, hydrogen desorption results from valence band excitations via a mechanism where one to two correlated secondary electrons directly excite electrons in the Si-H bond. We first quantify the hydrogen desorption as a function of photon irradiation using a high intensity, non-monochromatic synchrotron source and determine the desorption rate, yield, and cross-section of the desorption process. We then use a series of X-ray filters to limit the transmission bands of the non-monochromatic light while maintaining very high intensity to establish the EUV photon energy range responsible for the observed hydrogen desorption. Finally, we use monochromatic EUV photon exposures at 93 and 106 eV and demonstrate hydrogen desorption from the monohydride Si(001):H surface. Our results provide quantitative detail on hydrogen desorption from Si(001):H using photons in the EUV energy range and conclusively demonstrate that monochromatic EUV photons, with flux densities of order 100 W/cm$^2$, can usefully desorb hydrogen from this technologically relevant surface.

## Results and discussion

### Characterizing the clean Si 2$p$ photoelectron spectrum

We prepared clean and monohydride-terminated Si(001) surfaces in UHV (see Methods). Figure 1b, e shows, respectively, in situ XPS and STM measurements taken from an atomically clean Si(001) surface. This photoelectron spectrum is in good agreement with previous measurements:[30,31] the bulk silicon atoms contribute a pair of spin-orbit split peaks separated by 0.6 eV with a 1:2 peak height ratio, labelled B in Fig. 1b. The pristine (001)-c(4 × 2) surface reconstruction consists of rows of dimerised silicon atoms. These dimers undergo a Jahn-Teller distortion such that one atom is buckled up and the other buckled down, with a corresponding charge transfer from the down to the up buckled atom[32]. We label these contributions to the XPS as $S_{up}$ and $S_{down}$, respectively. The reconstruction of the surface-layer atoms has a knock-on effect to the first and second sub-surface ($SS$) layer atoms, producing additional sets of XPS peaks that we label $SS_1$ and $SS_2$. The STM image of our clean Si(001) surface shown in Fig. 1e confirms that we have an atomically-clean low defect density surface.

Figure 1c shows a Si 2$p$ photoelectron spectrum taken for a monohydride Si(001):H surface. The monohydride termination removes the surface dimer buckling and produces symmetric dimers, changing the reconstruction from c(4 × 2) to (2 × 1)[33,34]. Correspondingly, the XPS spectrum from this surface is simplified compared to the clean surface spectrum. We fit our spectrum with two sets of spin-orbit split peaks that are shifted to a lower binding energy by −250 and −450 meV with respect to the bulk component. The −250 meV shifted peak has the largest area and is equal in intensity to the sum of the two surface peaks ($S_{up}$ and $S_{down}$) from the clean surface. Thus, we assign this peak to the surface layer silicon atoms of the monohydride surface and label it as $S_H$. The negative binding energy shift of this component can be understood since hydrogen is more electronegative than silicon ($\chi_H$ (2.1) > $\chi_{Si}$ (1.8)[35]). The second set of shifted peaks (denoted as $SS_H$) is much lower in intensity and we attribute this to relaxations of the second layer, whose atomic displacements are approximately the same in magnitude as the clean Si(001) surface[34]. Using STM, we further confirmed the quality of our Si(001):H surface. Figure 1f shows one such image, confirming the surface cleanliness and low defect and dangling bond densities.

### Non-monochromatic irradiation of Si(001):H

As a first step to establishing EUV hydrogen desorption from Si(001):H, we irradiated a monohydride Si(001):H surface with very high intensity, non-monochromatic light generated by the zero-order full spectral distribution of the bending magnet at the PEARL beamline (see Methods). Figure 1d shows an XPS spectrum obtained after 100 min exposure, and Fig. 1g shows an atomic-resolution STM image taken over the same irradiated area.

For the irradiated Si(001):H surface, the XPS spectrum in Fig. 1d cannot be fit using only the peak components of the clean surface or the monohydride surface alone. Instead, the spectrum exhibits characteristics of both the clean and monohydride surfaces and is best fit with a combination of these peak components, reflecting the changing chemistry of the surface due to hydrogen desorption. We find a strong $S_H$ component attributable to hydrogen-terminated surface silicon atoms and strong $S_{up}$ and $SS_2$ components characteristic of clean silicon surface atoms.

Similarly, the STM data in Fig. 1g exhibit a mixture of clean and hydrogen-terminated silicon, where the clean silicon atoms appear bright against a minority background of hydrogen-terminated silicon surface atoms that appear darker due to having a lower density of states near the Fermi level[36]. We quantify the surface density of clean silicon in terms of monolayers (MLs; where 1 ML = 6.78 × 10$^{14}$ atoms cm$^{-2}$) by using an image threshold selection method[37] and find it to be 0.55 ± 0.10 ML for the STM data shown in Fig. 1g (see Supplementary Note 1 for more details). At saturation phosphine coverage, approximately one in four silicon (001) surface atoms are replaced by phosphorous during the silicon δ-layer growth[38]. This is because the optimal configuration for phosphorus incorporation is 3 neighbouring clean silicon dimers along a row[11,39]. Although the stochastic dopant incorporation percentage is not yet known for this technique, exposing a 0.55 ML equivalent of clean silicon to phosphine can be expected to produce a maximum of 0.14 ML surface density of phosphorus, or roughly seven times greater than required for the metal-insulator transition (-0.02 ML)[40]. If we assume that all 3-dimer sites are available for phosphorous incorporation in areas desorbed with EUV light, then from our STM data we estimate an incorporation efficiency of 20 ± 10% for phosphorous; this corresponds to a dopant density of 0.11 ML. Thus, this method of photon-based hydrogen desorption lithography can potentially create metallic, large-scale contacts or interconnects to multi-layer quantum devices[14,41]. We expect the incorporation and activation of dopants to be similar to what has been previously demonstrated on both clean surfaces[42,43] and hydrogen-terminated surfaces patterned with STM[44,45]. The only anticipated difference lies in the density of incorporated dopants, which we predict to be slightly lower due to any incomplete desorption of hydrogen. We further note that our STM data also demonstrate that the surface remains atomically clean and free of spurious contamination during the desorption process, which is critical for device lithography.

To gain further insight into the hydrogen desorption mechanism, we performed time-dependent measurements, where we repeatedly exposed a monohydride surface to high intensity, non-monochromatic light in 10 min intervals up to a total irradiation time of 140 min, measuring the Si 2$p$ XPS spectrum after each 10 min exposure. Fits were made to all the measured XPS spectra and in Fig. 2a, we show how the intensity of each component changes as a function of the irradiation time. The data allow us to follow the evolution of the photoelectron spectrum as the surface transitions from monohydride Si(001):H to partially clean Si(001) as a result of the non-monochromatic irradiation. We observe three distinct changes with increasing exposure: (i) decrease of the $S_H$ component (red), associated with the symmetric monohydride dimer silicon atoms; (ii) emergence of a component with a chemical shift of 500 ± 20 meV that can be attributed to the $S_{up}$ component (green) of the clean silicon surface; (iii) emergence of a component at 260 ± 20 meV which we identify as the $SS_2$ component (purple) of the clean Si(001) surface. Interestingly, a component with an energy shift of −450 meV remains constant throughout the exposure. We attribute this peak to the second layer silicon atoms of either the monohydride or clean surfaces (i.e., $SS_H$ or $SS_1$). We repeated the experiment for a substrate cooled to 77 K (both during desorption and measurement stages) to check for any temperature dependence of the hydrogen desorption. The low temperature data, plotted as open symbols in Fig. 2a, overlap with the

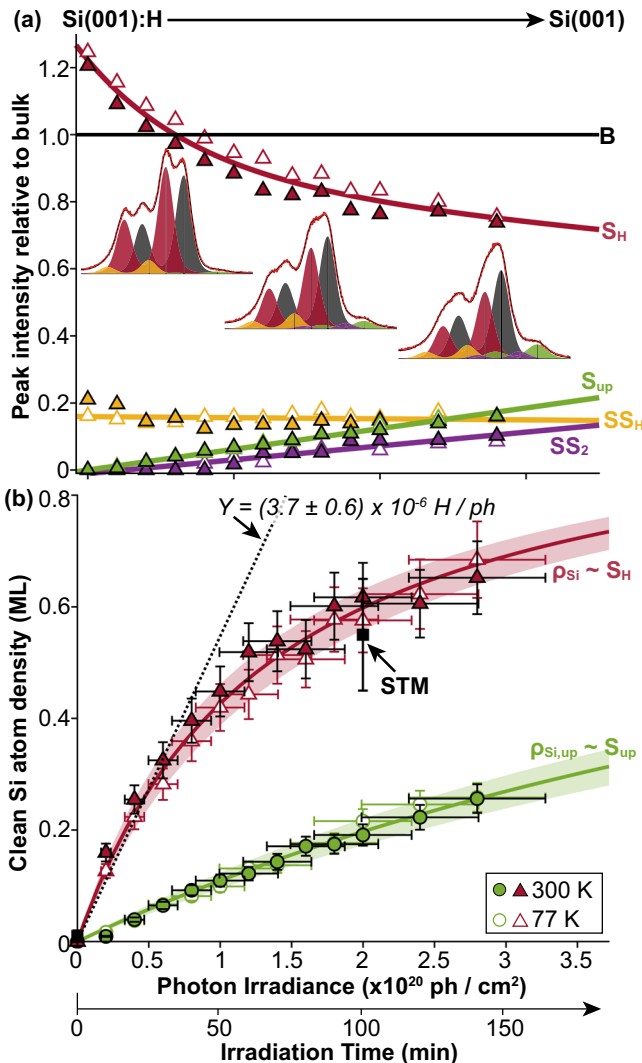

**Fig. 2 | Estimated clean silicon atom density versus non-monochromatic photon irradiance. a** Plot of the intensities of the Si 2*p* peak components (labelled and colour-coded) versus the non-monochromatic irradiation time/irradiance (see Methods for the definition of the photon irradiance). The bulk (*B*) peak intensity is normalised to one and the solid lines are guides to the eye. The inset XPS data at 20, 80 and 120 min illustrate the change of the Si 2*p* spectrum with hydrogen desorption. The XPS fits at each 20 min interval are shown in Supplementary Fig. 2. **b** Density of clean silicon atoms, $\rho_{Si}$ (red triangles) and up-buckled silicon atoms, $\rho_{Si,up}$ (green circles) versus non-monochromatic irradiation time/irradiance. The black data point at 100 min irradiation time shows the silicon atom density, as measured from the STM image. Data were taken at both room temperature (filled symbols) and 77 K (empty symbols), where the temperature was kept the same during both the desorption and measuring steps. The coloured areas around the fits represent the 90% confidence interval. In the linear regime, the desorption yield, *Y*, is equal to the gradient of the dotted black line. The experimental data and fits (solid lines) are performed over the domain $0 – 2.75 \times 10^{20}$ ph/cm² (or 0 – 140 min) using a one-parameter fit in accordance with Eq. 3. For $\rho_{Si}$, the best fit yields $\sigma = (0.7 \pm 0.1) \times 10^{-20}$ cm² and $v_{des.} = (0.015 \pm 0.001)$ ML/min for the photon irradiance and irradiation time domain, respectively. The vertical error bars of the STM were determined from repeated measurements at different locations on the surface, whereas the error bars from the XPS data was ~10%, derived from the curve fits. The horizontal error bars for the time domain are ~ 1 min, however, this is higher when transformed into photon irradiance due to the uncertainty of the photon flux and spot-size.

room temperature results, showing that the desorption mechanism is temperature-independent (see Supplementary Note 2, which provides a more detailed summary of all the fits for both room temperature (300 K) and at 77 K).

## Quantifying the desorption rate, yield, and cross-section
As discussed above, Fig. 2a shows that the $S_H$ component decreases as a function of non-monochromatic irradiation time due to the removal of hydrogen and hence the increasing density of clean (unpassivated) silicon atoms on the surface. We quantify the total density of such silicon atoms, $\rho_{Si}$, in units of MLs as a function of the photon irradiation time, *t*, when measuring the Si 2*p* photoelectron spectrum at a constant emission angle, by:

$$\rho_{Si}(t) = \frac{|S_H(t) - S_H(0)|}{S_{down}} \quad (1)$$

where $S_H(t)$ is the $S_H$ peak height (red component in Fig. 1d) relative to the bulk component peak height after an irradiation time *t*, and $S_{down}$ is the $S_{down}$ peak height (red component in Fig. 1b) relative to the bulk component peak, which is used as a reference for a clean silicon surface. Thus, Eq. 1 essentially measures the percentage difference of the $S_H$ component of the irradiated Si(001):H surface relative to the known value of the $S_{down}$ peak for the clean Si(001) surface, i.e., the percentage decrease in the $S_H$ component that goes into generating a percentage increase in clean silicon atoms. Conversely, 1 - $\rho_{Si}(t)$ represents the surface density of hydrogen since there is a one-to-one ratio of silicon to hydrogen for the monohydride surface.

To establish the mechanism for the observed hydrogen desorption, we must quantify the desorption rate ($v_{des.}$), yield (*Y*), and cross-section ($\sigma$). To achieve this, we first define the photon irradiance ($\Sigma_{ph}$), i.e., the total number of photons received by the Si(001):H surface per unit area (see Methods). Figure 2b shows how the total density of clean silicon atoms, $\rho_{Si}$, varies as a function of the photon irradiance (or irradiation time). Here, we also see that the $\rho_{Si}$ fit (red curve) agrees with the STM data point where the dangling bond density was directly measured. This is the first demonstration that XPS can be used to directly measure the dangling bond density of a monohydride Si(001):H surface via the $S_H$ photoelectron component.

Using a similar approach, the total number of up-buckled unpassivated silicon atoms can also be determined, based on the measurement of the $S_{up}$ component. In Fig. 2a, this component initially starts at zero for the Si(001):H surface, so we write the corresponding density of up-buckled unpassivated silicon atoms as:

$$\rho_{Si, up}(t) = \frac{S_{up}(t)}{S_{up}} \quad (2)$$

which measures the ratio of the $S_{up}$ peak height for the irradiated Si(001):H surface (the green component in Fig. 1d) after a time *t*, to the $S_{up}$ peak of the clean Si(001) surface (the green component in Fig. 1b); both are measured relative to their corresponding bulk component peak heights and are plotted in Fig. 2b as the green curve. Since this does not count the dimers that remain symmetric after desorption (due to strain or nearby defects), Eq. 2 leads to an underestimation of the desorbed hydrogen. Thus, when determining the total dangling bond density of the surface, we use Eq. 1 and the $S_H$ component of the XPS spectra.

The most important aspect of Fig. 2b is that the values of the clean silicon atom density appear to vary non-linearly and approach 1 ML at an ever-decreasing rate. Furthermore, the desorption rate is found to be independent of the substrate temperature and a 1-parameter fit is sufficient to model the data. The functional fit to the data shown in Fig. 2b is

$$\rho_{Si}\left(\Sigma_{ph}\right) = \frac{1}{1 + \frac{1}{\sigma \Sigma_{ph}}} \quad (3)$$

where $\Sigma_{ph}$ is the photon irradiance and $\sigma$ is the rate constant (equal to the desorption cross-section). For non-monochromatic irradiation, we find that the desorption cross-section is $\sigma = (0.7 \pm 0.1) \times 10^{-20}$ cm². This

value is approximately 100 times smaller than the average photo-ionization cross-section for silicon in the EUV energy range, and 10 times smaller than the hydrogen cross-section[46]. Since the $y$-axis in Fig. 2b is the clean silicon atom density, the gradient in the initial linear regime (dotted line in Fig. 2b) gives an estimate of the desorption yield, $Y = (3.7 \pm 0.6) \times 10^{-5}$ ph$^{-1}$. Equation 3 can be equivalently written in terms of the irradiation time, where the rate constant then becomes the desorption rate, $v_{des} = (0.015 \pm 0.001)$ ML/min. The values for the desorption cross-section and yield lie within the anticipated range for hydrogen desorption found previously in the literature for electron-[16,17,47] or photon-stimulated[22,23,48], desorption.

Equation 3 is an exact solution of the first-order non-linear differential equation:

$$\frac{d\rho_{Si}}{d\Sigma_{ph}} = \frac{1}{\sigma}\left(1 - \rho_{Si}\right)^2 \tag{4}$$

In Eq. 4, the population density, $\rho_{Si}$, of bare silicon atoms is supplied by hydrogen passivated atoms at a rate that is proportional to the surface density of hydrogen, equivalent to $1 - \rho_{Si}$, implying that as the hydrogen desorption progresses, the probability of further desorption decreases, leading to the non-linear $(1 - \rho_{Si})^2$ term and an overall slowing down of the desorption (see Supplementary Note 3 for plots comparing the model solution and its derivative). The non-linear nature of the desorption probability and the fact that it depends strongly on the remaining surface coverage of hydrogen suggests an electron-stimulated mechanism, where electrons emitted from photo-stimulated neighbours trigger the desorption of hydrogen; this is argued later with further evidence.

Overall, the clean silicon atom density gradually approaches 1 ML and to achieve a clean silicon atom density of 0.95 ML, a photon irradiance of $12 \times 10^{20}$ ph/cm$^2$ is needed – equivalent to 18,000 Joules/cm$^2$ or a three-minute exposure with an EUV (13.5 nm) intensity of 100 W/cm$^2$. For comparison, modern EUV lithography sources have a power of 300 W at the intermediate focus, however, this power is significantly reduced to approximately 6 W at the wafer level due to the reflectivity loss of the optics[49]. Expectations for future systems predict a substantial increase of power at the wafer level, with predictions reaching up to 800 W for EUV sources and a reduced number of mirrors[50]. Therefore, a throughput of minutes per chip can be achieved. While this throughput might seem low compared to the production of classical devices, it is crucial to note that the scaling of computing power for quantum devices is exponential, unlike classical computers, which scale linearly. This means that quantum devices require significantly fewer qubits to surpass the performance of classical transistors.

## Establishing the photon energy range of hydrogen desorption

To investigate which photon energies are responsible for the observed hydrogen desorption, we inserted three separate X-ray filters into the beamline, after the refocussing mirror. The three filters used were 250 nm thin films of Al, Zr, and C (see Methods for filter specification details). We show a plot of the calculated and measured transmittance for each filter in Fig. 3c. The data points are measured via the sample drain current using a bare silicon substrate for photon energies between 100–1000 eV and we see good agreement with the calculated curves. Therefore, using each one of these X-ray filters, we can access different transmission bands in the EUV range and, importantly, filter photons with energies < 10 eV from the non-monochromatic spectrum to rule out any direct photon-induced desorption of Si-H.

Figure 3a shows the Si 2$p$ photoelectron spectrum of the Si(001):H surface after 80 min non-monochromatic irradiation using each of the three filters, and one for the unfiltered beam. As expected, the hydrogen desorption leads to a decrease in the hydrogenic $S_H$ component and an increase in the silicon $S_{up}$ component (inset of Fig. 3a). Since the irradiation time was the same for all filters, the differences in

the photoelectron spectra derive from the different transmitted spectral components. We measure the relative change in the desorption rate by monitoring the $S_{up}$ peak intensity, which is associated with the clean (unpassivated) silicon atoms. The $S_{up}$ component was selected because it appears clearly in the photoelectron spectrum without interference from other peaks. This allows for a precise measurement of the relative change in the desorption rate using each X-ray filter. Examining the spectra in Fig. 3a reveals that all three filters result in a reduction in hydrogen desorption, with the Al filter having the greatest impact and the C filter having the least. Figure 3b provides a closer look at how the $S_{up}$ component changes as a function of the irradiation time (see Supplementary Fig. 3a for the corresponding photon irradiance plots). These data demonstrate that the Al filter reduces the desorption almost to zero (2% relative to the unfiltered exposure), and the Zr filter also heavily attenuates the desorption (16%). In contrast, hydrogen desorption remains relatively strong when using the C filter (69% as efficient as unfiltered).

Comparing the relative desorption rates for the three filters to their transmission curves in Fig. 3c allows us to determine the most likely range of photon energies that are responsible for the hydrogen desorption. In particular, the relative ordering of the transmission for photon energies between 100 – 210 eV (Al < Zr < C) matches our experimental hydrogen desorption data, which therefore puts a bound on the photon energies responsible for hydrogen desorption (shown by the green area in Fig. 3c). Since the Si 2$p$ and Si 2$s$ core-levels lie within this energy range, this suggests that the ionisation of these core-levels may play a role in the observed hydrogen desorption. However, we show later that this is due to the increased yield of secondary electrons generated as a by-product of exciting these core-levels; for example, whether the photons are above or below the Si 2$p$ excitation, the desorption persists, but, at a smaller rate.

K-shell Auger relaxation mechanisms require photon energies > 1840 eV where the transmission for all filters tends to 1. Hence, if such Auger processes were responsible for the observed hydrogen desorption, we would expect the relative desorption rates in Fig. 3b to be identical for each filter; this is contrary to what we observe. Furthermore, for photon energies < 15 eV, all three filters have an effective transmission < 10$^{-6}$. This indicates that the 7.9 eV photons are also not responsible for the observed hydrogen desorption since we would expect the relative desorption rates in Fig. 3b to be all near zero. These measurements establish not only that photons in the EUV range can initiate hydrogen desorption, but they also rule out two possible desorption mechanisms; K-shell Auger relaxations[26] and the 7.9 eV photons[21-23] that are known to cause hydrogen desorption from Si(001). These findings point towards a mechanism mediated by secondary electrons.

## Monochromatic irradiation of Si(001):H

For technological applications of photon-based hydrogen desorption lithography, desorption via monochromatic irradiation is important for compatibility with existing semiconductor processing methods. In addition, using monochromatic irradiation enables interference-based patterning, where multiple coherent beams are made to interfere constructively and destructively to facilitate a large-scale, high-resolution patterning of the resist[51].

From our non-monochromatic exposures presented above it is clear that (i) the observed hydrogen desorption is not induced by the photons directly and (ii) very high beam intensities are necessary to induce appreciable desorption. Therefore, for monochromatic exposures, we chose a photon energy of 109 eV, which is 10 eV above the Si 2$p$ core-level where we expect a maximal generation of photoelectrons, to maximise the probability of hydrogen desorption.

Figure 3d shows Si 2$p$ photoelectron spectra of the monohydride Si(001):H surface, taken before and after $0.05 \times 10^{20}$ ph cm$^{-2}$ (or 900 min), 109 eV monochromatic photon irradiation. The inset shows

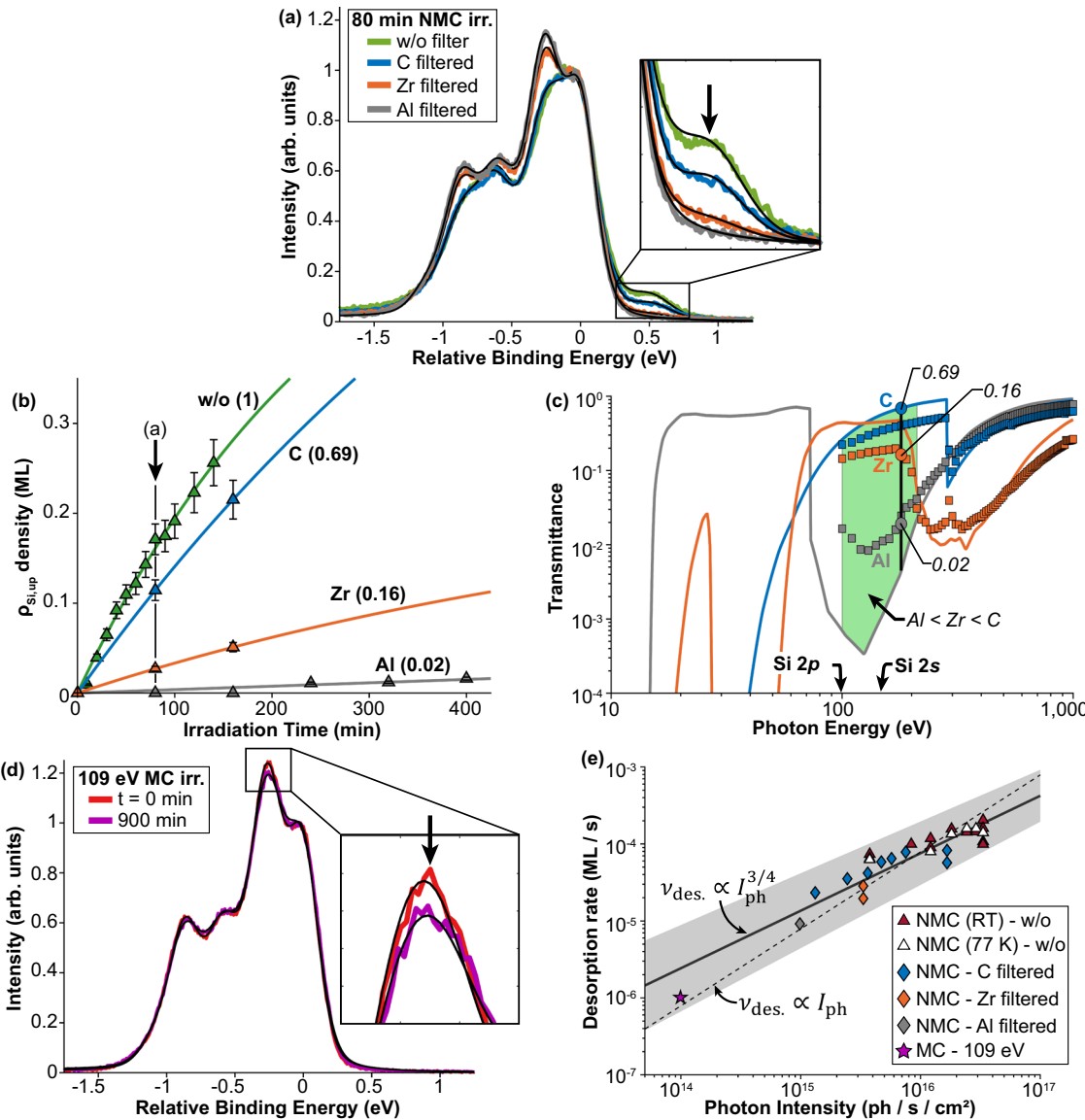

**Fig. 3 | Extreme ultraviolet (EUV) photon energy bandwidth for hydrogen desorption. a** Si 2$p$ photoelectron spectrum taken after 80 min photon irradiation using non-monochromatic (NMC) light without (w/o) and with three different EUV filters from a Si(001):H surface. A zoom-in of the single dangling bond component is shown by the arrow. **b** Plot of the density of up-buckled silicon atoms, $\rho_{Si,up}$, as a function of photon irradiation time (symbols), along with their best fits (lines). Each filter is labelled along with its comparative desorption rate relative to its unfiltered value. **c** Plot of the calculated (solid lines) and experimental (symbols) transmission curves for each EUV filter. The green region spans the photon energy range 100–210 eV where the transmission profile is such that Al < Zr < C; this coincides with the observed desorption rates in (**a**, **b**) and represents the photon energy

regime responsible for the hydrogen desorption. The arrows indicate the threshold energy of the Si 2$p$ and Si 2$s$ core-levels, which lie within the green region. **d** Si 2$p$ photoelectron spectrum taken after 900 min photon irradiation using mono-chromatic (MC) light at 109 eV. A zoom-in of the hydrogenic component is shown; the arrow indicates a very small decrease, due to hydrogen desorption. **e** Log-log plot of the desorption rate, $\nu_{des.}$, as a function of incident photon intensity, $I_{ph}$. The shaded area around the fit represents the 90% confidence interval. The NMC exposures are shown for both 77 K and RT, whereas the filtered NMC and MC exposures are at RT. The solid and dashed lines show the best fit for an exponent that equals 3/4 (sub-linear) and 1 (linear), respectively.

an enlargement of the peak where a small decrease in the amplitude of the $S_H$ component is evident, corresponding to a created dangling bond density of 0.05 ML, estimated using Eq. 1. Although the exposure time was long, the observed desorption is modest due to the relatively low incident flux achievable using a monochromatic irradiation at the PEARL beamline (2 orders of magnitude smaller than the zero-order beam used for the non-monochromatic exposures above). This discovery is significant as it shows for the first time that monochromatic EUV light can desorb hydrogen on the monohydride Si(001):H surface. With a high enough flux, this has the potential to be highly effective, as demonstrated in our XPEEM experiments in the next section. A review of all exposures performed at PEARL can be found in Supplementary

Note 3; all data points of clean silicon atom density versus photon irradiance are found to lie in very good agreement with one another.

Figure 3e summarises all our results for the measured desorption rate as a function of the incident photon intensity (flux per unit area). Here, the photon intensity was controlled by reducing the size of the front-end aperture of the beamline, which was done in discrete steps for the unfiltered and C filtered non-monochromatic irradiations. A good fit to the data is provided by a power-law:

$$\nu_{des.} = A\left(\frac{I_{ph}}{I_0}\right)^{3/4} \tag{5}$$

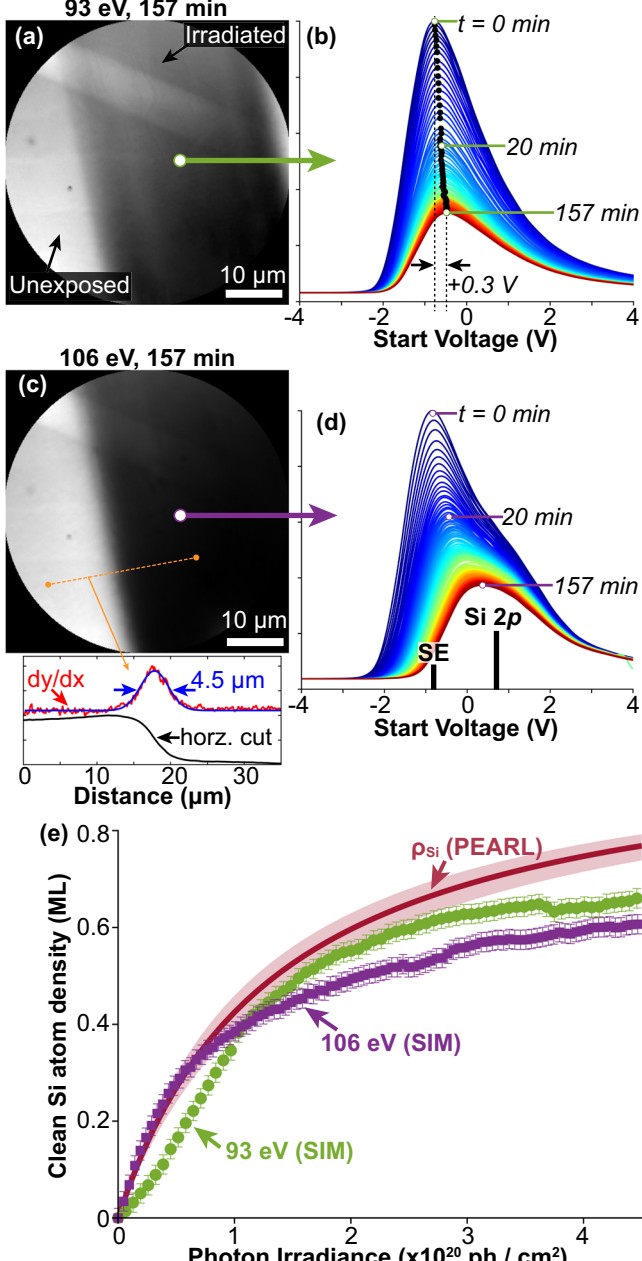

**Fig. 4 | Demonstrating hydrogen desorption in the EUV range using XPEEM at the SIM beamline. a, c** XPEEM images taken with a start voltage of −0.8 V and 50 μm field-of-view after a (**a**) 93 eV and (**c**) 106 eV irradiation for 157 min. A vertical, rectangular band was irradiated by narrowing the beamline exit slit to be smaller than the PEEM field-of-view, after which it was widened to reveal both the unexposed and irradiated areas (see Methods for more details). **b, d** A cumulative series of secondary electron (SE) curves, extracted every 1 min during the (**b**) 93 eV and (**d**) 106 eV irradiations. In (**b**), a shift of the SE peak of +0.3 V is observed. In (**d**) the vertical, black, solid lines indicate the SE peak position (at 0 min) and the Si 2*p* photoelectron peak position. In (**c**), the inset at the bottom shows a line profile extracted from the XPEEM image (denoted by the yellow, dashed line), with the measured FWHM of the edge being 4.5 μm. **e** Plot of the density of clean silicon atoms as a function of the photon irradiance. The purple and green curves show the results from the SIM beamline irradiation and the red curve is the best fit made to the PEARL experiments of Fig. 2b. Note that the presence of the Si 2*p* shoulder in (**d**) can obfuscate the true SE peak height, causing it to be underestimated; this is what causes the purple curve in (**e**) to deviate from the PEARL curve. This is absent in (**b**), where only a single SE peak is observed, since 93 eV is below the Si 2*p* excitation threshold. In (**e**), the vertical error-bars are -0.02 ML.

where $v_{des.}$) is the desorption rate, $I_{ph}$ is the incident photon intensity, $I_0$ is a normalisation constant which we set to the unfiltered non-monochromatic photon intensity and $A$ is a fit constant equal to $8.5 \times 10^{-17}$ ML s$^{-1}$ cm$^{-2}$. The exponent which is 3/4 in Eq. 5 provides the best description of the data over the three decades of photon intensities, but an exponent of 1 (linear behaviour, dashed line in Fig. 3e) is not excluded given the errors in our measurements. We note also that since Eq. 5 is phenomenological, it could be replaced by a function which is linear up to $10^{16}$ ph/s/cm$^2$ (dashed line in Fig. 3e) and then sublinear (solid line in Fig. 3e).

For Auger relaxation and direct photoexcitation desorption, we expect the desorption probability to be linear with the photon flux until the usual non-linearities associated with strong photon fields set in. A very conservative estimate as to the relevance of the latter follows from considering that a beam of $10^{20}$ ph/s/cm$^2$ will result in the passage of roughly $10^5$ ph/s through each Si atom. This is much smaller than both the 500 MHz pulse repetition rate of the synchrotron and the relaxation rate of any core level, meaning that non-linear quantum optics effects (analogous to those reported in[52] and references therein) can be ignored. Thus, assuming the statistical preference for a desorption rate which becomes sublinear for photon intensities more than $10^{16}$ ph/s/cm$^2$, we have further confirmation that it is the secondary electrons that are responsible for the Si-H bond scission. In particular, the desorption process becomes less efficient as the photon intensity increases, not because of multiphoton absorption, but because the absorption probabilities for secondary electrons are reduced after one secondary electron has already interacted with a Si-H bond.

**XPEEM imaging of Si(001):H using monochromatic irradiation**

As a proof-of-principle demonstration that spatial patterning is possible via EUV hydrogen desorption, we have completed measurements of XPEEM at the endstation of the SIM beamline of the SLS. This undulator beamline provides a monochromatic flux two orders of magnitude higher than the PEARL beamline (see Methods), and, therefore, provides monochromatic light of similar intensity to the non-monochromatic, zero-order beam discussed above.

With XPEEM, photoelectrons are excited with an incident X-ray photon beam and the local intensity of the energy filtered photoelectrons reaching the detector is measured. The PEEM energy analyser allows us to probe electrons with a specific kinetic energy, selected via the start voltage (SV), applied between the analyser and sample. For secondary electron spectroscopy, the SV is typically varied between −5 to 5 V (see Methods for more details).

In two separate experiments, we exposed a Si(001):H surface to EUV light at 93 and 106 eV, respectively. First, the exit slit of the X-ray optics was narrowed so that only a 25 μm vertical, rectangular band of the sample surface was irradiated. After 157 min, the exit slit was widened to span the entire XPEEM field-of-view and Fig. 4a, c show the resulting XPEEM images taken after these exposures. The XPEEM images clearly show a contrast difference between the unexposed and exposed surfaces, with the exposed surface appearing darker in contrast. Additionally, the 106 eV exposed area in Fig. 4c is darker in contrast compared to the 93 eV exposed area in Fig. 4a. The implication of the observations is that both irradiations have induced hydrogen desorption, and that the desorption is more effective at 106 eV compared to 93 eV; we clarify this below.

The contrast observed in the XPEEM images (Fig. 4a, c) is attributed to a change in the work function induced by hydrogen desorption within the exposed region of the sample surface. The work function of bare silicon is $\phi_{Si} = 4.80 \pm 0.10$ eV[53] while that of hydrogen terminated silicon is $\phi_{SiH} = 4.24 \pm 0.04$ eV[54]. Since the work function of hydrogen-terminated silicon is lower, this supports the polarity of the observed contrast. The higher work function of bare silicon makes it less likely for low energy electrons to escape, resulting in a lower intensity of

photoemitted electrons (dark XPEEM contrast) compared to areas that remain hydrogen terminated (bright XPEEM contrast). We rule out impurity deposition (for example, from the cracking of residual hydrocarbon species) as an alternative explanation of the contrast difference since the chamber is under an UHV ($< 5 \times 10^{-10}$ mbar) and the timescale of the X-ray exposure was only several minutes.

During the monochromatic irradiation, we also recorded the secondary electron curves by sweeping the PEEM start voltage from −4 V to 4 V over a region of the XPEEM image that was actively being irradiated. Each of these measurements took 1 min and Fig. 4b, d show these series of curves for the 93 and 106 eV irradiations, respectively. Every curve shows a peak in the secondary electron emission spectra, and we see that these shift to a more negative voltage as the exposure progresses. By measuring the difference in the peak position at $t = 0$ min and 157 min in Fig. 4b, we can determine that the work function increases by ~0.3 eV during the exposure, consistent with hydrogen desorption; since $\Delta\phi = \phi_{Si} - \phi_{SiH} = 0.56 \pm 0.14$ eV, this aligns well with the observed increase in start voltage shift, however, the magnitude of this shift is smaller than anticipated, likely indicating that saturation desorption has not been fully achieved yet. Similar behaviour is seen for the 106 eV irradiation; however, Fig. 4d has a second peak on the right-hand shoulder of the secondary electron curve; this is due to the additional low kinetic energy photoelectrons that originate from the Si 2$p$ core-level excitations. Since the total area beneath the curves in Fig. 4b, d is proportional to the total number of low kinetic energy electrons emitted from the sample surface, the desorption rate for the 106 eV irradiations should be higher than the 93 eV exposures. This explains the origin of the darker contrast in their respective XPEEM images (Fig. 4a, c), since the 106 eV exposure leads to more hydrogen desorption, and indicates that the secondary electrons are mediating the hydrogen desorption, consistent with the PEARL XPS measurements.

To quantify the desorption rate, we use a similar model underlying Eq. 1, namely, we measure one minus the relative peak height of the secondary electron curve as a function of time, such that:

$$\rho_{Si}(t) = 1 - \frac{I(t)}{I(0)} \qquad (6)$$

where $I(t)$ is the peak height of the secondary electron curve at a time $t$ and $I(0)$ is the initial peak height at $t = 0$. Figure 4e shows the result of applying Eq. 6 to Fig. 4b, d. We find that the irradiations performed at 93 eV and 106 eV in our XPEEM experiment at the SIM beamline align closely with the best fit of the hydrogen desorption curve that was measured in our monochromatic and non-monochromatic exposures at the PEARL beamline (as shown in Fig. 4e). This confirms that the total dangling bond density measured at the PEARL and SIM beamline are consistent. The agreement between the measurements made independently at the PEARL and SIM beamlines further supports the interpretation of hydrogen desorption using monochromatic irradiation. Furthermore, we find that the desorption process scales with secondary electron production, measured directly by the total electron yield, and which has resonant enhancements whenever new core-levels become available. Since there is little difference between the 93 and 106 eV irradiations, this suggests that valence band excitations are the prevalent source of secondary electrons mediating the desorption process.

## Hydrogen desorption mechanism

Our results indicate that the principal desorption mechanism is mediated by secondary electrons. There are two candidate mechanisms by which electrons can initiate hydrogen desorption. The first is associated with a single excitation process, whereby a single secondary electron with a kinetic energy of ≈6.5 eV promotes an electron from the $\sigma$ orbital into the $\sigma^*$ orbital of the Si-H bond, where the subsequent repulsion of the hydrogen atom gains sufficient kinetic energy to

desorb. The second is associated with a multiple excitation process that is mediated via the generation of a two core-hole final state localised within the Si-H bond[55]. Here, the core-holes are generated by the secondary electrons as they interact with the Si-H bond. These electrons will have typical kinetic energies between 1 – 5 eV. Since the rate of reneutralisation of a one-hole state is predicted to lie between 0.1 – 10 fs, this means the excitations are expected to be simultaneous[55]. Thus, for this mechanism, hydrogen desorption occurs via the simultaneous generation of a two core-hole final state in the Si-H bond, where the bonding electrons are either ejected from the surface, excited into an unoccupied state of the conduction band or relaxed via a valence band Auger decay. Generally, both desorption mechanisms may well run in parallel and, depending on the kinetic energy of the secondary electrons, either one or two electrons are required to initiate Si-H bonding scission.

A silicon dimer with only one hydrogen attached (Si-Si-H) is called a hemi-hydride dimer, whereas a silicon dimer with two hydrogens attached (H-Si-Si-H) is called a monohydride dimer. The energy per silicon atom required to desorb a hydrogen atom from the surface in either configuration (hemi- or mono-hydride) is roughly the same (~3.7 eV), with the difference between them of just 78 meV[56]. Thus, we infer that both configurations of adsorbed hydrogen have an equal likelihood to produce hydrogen desorption under EUV illumination. This results in a surface that exhibits a random distribution of clean Si dimers, as seen in the STM data in Fig. 1g. However, as stated above, our EUV desorbed regions have sufficient 3-dimer sites for the patterned areas to achieve an incorporation yield higher than that required for the metal-insulator transition (~0.02 ML)[40].

Since we have established that the hydrogen desorption mechanism is mediated by secondary electrons, this offers an explanation to the origin of the mitigated desorption probability as the hydrogen desorption progresses. As hydrogen is desorbed, the local work function increases, leading to a decrease in the number of secondary electrons being ejected per unit time to interact with a Si-H bond. This, in turn, decreases the probability of hydrogen desorption, explaining the observed decrease in the hydrogen desorption rate with irradiation time (Fig. 2b). The change in the local work function is evident in the contrast difference in the XPEEM images of Fig. 4a,c and the decrease in the area beneath the secondary electron curves of Fig. 4b, d confirms that fewer secondary electrons are emitted from the surface for longer irradiation times. This asymptotic process causes the clean silicon atom density to gradually approach 1 ML, with an ever-decreasing desorption rate, consistent with Eq. 4. However, by using a higher photon flux, the desorption can be achieved more quickly, as the total number of secondary electrons is higher, as evidenced by the increase of the desorption rate with photon flux (Fig. 3e).

## Resolution of EUV-based hydrogen desorption lithography

Since the principal desorption mechanism is associated with electron interactions, we can make some arguments on the spatial resolution of the hydrogen desorption. The full width at half maximum (FWHM) across the desorption edge is measured from the XPEEM image in Fig. 4c and found to be 4.5 μm. We attribute this large line width to the divergence of the monochromatic beam through the exit slit and suboptimal focusing of the x-ray optics. We expect the ultimate resolution to be set by the interaction range of electrons with the hydrogen resist. In the first instance, this can be estimated from the universal inelastic mean free path (IMFP) curve[57]. For electrons with a kinetic energy ≈6.5 eV, an IMFP of 3.5 nm is found. This number is far better than achieved in previous studies of electron beam lithography on the monohydride Si(001):H surface, where incident electrons between 40 – 150 eV yielded a Gaussian width of 380 nm across the desorption edge[15]. Recent studies on EUV interference lithography using spin-coated photoresists have demonstrated high-resolution, sub-10 nm patterning[58,59]. Additionally, resistless photon-induced oxide

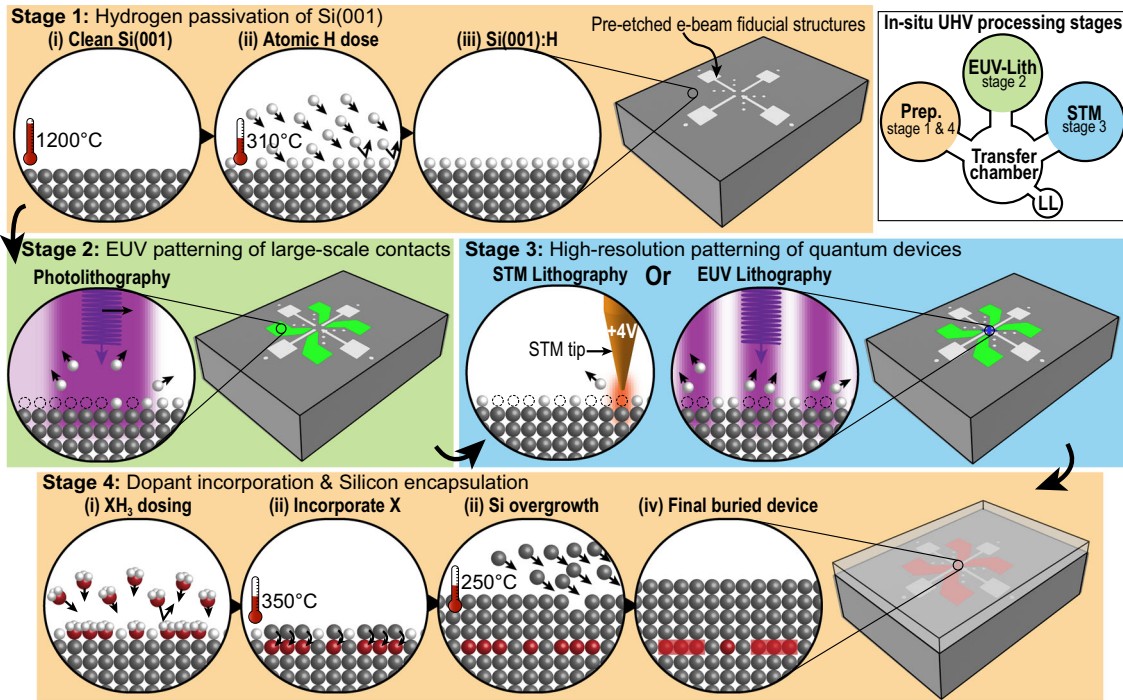

**Fig. 5 | Illustration of a proposed laboratory process flow of patterning silicon quantum devices, divided into four stages.** In the upper-right panel, we show the proposed UHV setup that can be integrated at the EUV-IL system at the XIL beamline of the SLS. The load-lock (LL) is where the silicon substrates are introduced into vacuum and then transferred to the transfer chamber. Three chambers are branched from the transfer chamber, all of which have a UHV environment with a base pressure $< 5 \times 10^{-10}$ mbar: the preparation, EUV-lithography and STM chamber. The preparation chamber is used for annealing, hydrogen passivation, $XH_3$ dosing and silicon MBE growth. The EUV-Lithography chamber is used for the EUV hydrogen desorption lithography. The STM chamber is used to gauge the quality of the surface at each stage and to perform atomic-scale hydrogen-desorption lithography. There are 4 main stages when it comes to fabricating silicon quantum devices; Stage 1: The process begins with preparing a clean silicon surface, followed by the atomic hydrogen passivation of Si(001). Stage 2: EUV hydrogen-desorption lithography of large-scale contacts. Stage 3: STM or EUV hydrogen-desorption lithography for patterning the nm-scale quantum device at the centre of the fiduciary markers. Stage 4: Dopant incorporation and silicon encapsulation. After this, the sample can be extracted from UHV into ambient and standard cleanroom processing techniques can be used to contact the buried device with vertical electrical connections[71].

patterning of silicon has a demonstrated half-pitch of just 75 nm[60]. Therefore, we expect that a spatial resolution of a few nanometres can be realistically achieved with EUV-based hydrogen desorption lithography. This is more than sufficient to fabricate δ-layer interconnects between the active components of quantum or classical devices, or to pattern localised dopant δ-layers in silicon. Furthermore, we expect that this method of photolithography can also be used to measure the spatial characteristics of an incident synchrotron beam for spot-profile diagnostics.

**Towards large-scale EUV patterning of silicon quantum devices**
To establish the compatibility and resolution of EUV-based hydrogen desorption lithography, we propose a laboratory setup where both EUV- and STM-based lithography are integrated. This integration aims to push the boundaries of what is currently achievable in the manufacture of silicon quantum devices. While this arrangement might initially seem complex and costlier than current fabrication tools, the EUV-IL system at the XIL beamline of the SLS already meets many of the required specifications, most importantly, an intense and spatially coherent beam of photons with energies of up to 500 eV already in routine use for sub-10 nm patterning[18,19]. However, XIL needs an upgrade given that it currently operates in the high vacuum regime (~$5 \times 10^{-7}$ mbar) and does not offer certain standard surface science tools: (i) Preparing Si(001):H requires a UHV system ($< 5 \times 10^{-10}$ mbar) to maintain surface cleanliness on the atomic-scale; (ii) An STM (as at PEARL) would be helpful for direct observation of the success of the EUV lithography and mixed STM / EUV lithography; (iii) An MBE chamber with the capability of dopant precursor gas dosing and epitaxial silicon overgrowth. With these upgrades, we foresee a laboratory with the ability to fabricate quantum devices in silicon using high-resolution EUV-based hydrogen desorption lithography. Our proposed laboratory process flow of patterning silicon quantum devices with EUV is encapsulated in Fig. 5.

Figure 5 shows the basic four stages for patterning silicon quantum devices (stages can be repeated for fabrication of structures in three-dimensions following[61]), and we foresee that two of these stages can be effectively implemented by EUV lithography. Our proposition extends beyond the use of STM for the patterning of quantum devices and encompasses the potential application of emerging EUV lithography techniques. For example, the patterning of nanodot-arrays using EUV achromatic Talbot lithography has achieved dot sizes of 20 nm[62,63] and we expect that it can be similarly used to fabricate devices that leverage the periodic arrangement of quantum dots to attain topological states in silicon[64]. Additionally, the periodic, mutually perpendicular (criss-cross) pattern of control gates proposed for a surface code quantum computer architecture in silicon[14] can also be feasibly patterned with EUV lithography. Thus, in contrast to STM, we believe that EUV lithography presents a viable alternative for achieving scalability.

In instances where there is a need to align EUV-patterned hydrogen structures with STM-patterned atomic structures, we propose an in situ top-down approach: starting with large-scale EUV patterning to draw large contact pads, followed by atomic-scale device patterning with the STM. This approach allows for local correction of any initial EUV pattern offset with the STM. Moreover, a proven method involves the use of pre-etched e-beam fiducial structures on the initial silicon

substrate that can withstand the UHV surface preparation[65,66] (an example of such markers is shown in Fig. 5). These can also serve as reference markers for establishing electrical connections to the buried and patterned nanostructures[66]. Nevertheless, the alignment of EUV-patterned hydrogen structures with buried STM-patterned atomic structures presents a significant challenge and is an ongoing area of research.

Our work, therefore, demonstrates the desorption of hydrogen from the Si(001):H surface using EUV light, thereby introducing a new, resistless EUV lithography method for nanoscale device patterning and dopant localization. This method is not only compatible with current STM-based fabrication, but also with commercial device patterning, which use similar photon energies. We show through XPS and STM measurements that hydrogen desorption occurs through non-monochromatic, filtered non-monochromatic, and monochromatic EUV exposures, and establish that the mechanism is photon-induced, temperature-independent, flux-dependent, and mediated by secondary electrons. We have also developed a method to quantify the dangling bond density with XPS and found that the clean silicon atom density extrapolates to 1, with an ever-decreasing desorption rate. XPEEM confirms the spatial patterning capability of this method. By combining with EUV interference-based patterning techniques, this approach offers a novel way to create nanoscale, spatially confined dopant patterns in silicon for use in nanoscale devices or as interconnects for atomically precise STM patterning. The implications for the semiconductor industry are significant, as EUV lithography using 92 eV (or 13.5 nm) is now widely used, whereas STM-based patterning remains exclusively in the domain of small-scale laboratory device demonstrations. The next step is to integrate this method with carrier gas and silicon molecular beam epitaxy systems. This will allow us to compare devices fabricated using photon-based lithography with those created using STM lithography, thereby confirming the compatibility of this technique.

## Methods

### Sample fabrication
Two types of samples were prepared: Si(001) and the monohydride Si(001)-(2 × 1):H surface. The starting point for both is an Sb-doped, $n$-type Si(001) substrate, with a miscut <0.01° and a resistivity of 0.05 Ω cm. The substrate was loaded into the PEARL preparation chamber, with a nominal base pressure < $10^{-10}$ mbar. Sample outgassing was carried out overnight (~12 h) at 600°C by passing a direct current through the sample, followed by flash annealing to 1200°C for 15 s. This desorbs the surface oxide and produces an atomically clean Si(001) surface, verified with in situ STM and XPS (Fig. 1b, e). The sample temperature was monitored using an infrared pyrometer (IMPAC IGA50-LO-plus), with an estimated uncertainty of ±30°C. Hydrogen termination was achieved through the use of an atomic hydrogen source (MBE Komponenten); the sample is held at a constant temperature of 330°C whilst dosing with atomic hydrogen at $5 \times 10^{-7}$ mbar for 5 min, which saturates the Si(001) surface with hydrogen. This induces the monohydride Si(001)-(2 × 1):H surface, which is again verified with in situ STM and XPS (Fig. 1c, f). At the SIM beamline, similar sample preparation methods were used.

### Combined XPS and STM at PEARL
All XPS measurements were performed using a Scienta EW4000 spectrometer, with an energy resolution $E_{pass}/\Delta E = 1750$. Both the beamline and spectrometer settings were consistent across the entire XPS datasets; a photon energy of 140 eV was used, a pass energy of 10 eV, integration time of 0.1 s, with an energy step size of 10 meV, with a total of 5 iterations. All STM experiments were performed within an Omicron low-temperature series STM at 77 K. In both cases, the measurements were carried out at a base pressure < $5 \times 10^{-10}$ mbar.

### XPS fitting procedures
A least-squares fitting procedure was implemented, in which the spectra were deconvolved into a series of components consisting of spin-orbit split Voigt functions, after background subtraction with a Shirley function[67]. All fits are performed using the IgorPro FitXPS package. An energy resolution of 0.05 eV is also set by the beamline optics and photoelectron analyser. For the Si 2p component fittings, the spin-orbit splitting was set to $0.60 \pm 0.02$ eV, where the peak intensity of the $2p_{1/2}$ component was set to 0.50 ± 0.02 of the $2p_{3/2}$ component, consistent with the literature value of the branching ratio. For the Voigt function, the Lorentzian linewidth was $0.06 \pm 0.01$ eV, with a Gaussian linewidth of $0.28 \pm 0.03$ eV. These parameters were consistent across all fits to the XPS data.

### Non-monochromatic photon irradiation
Non-monochromatic irradiation was performed at the PEARL beamline (featuring a bending magnet) at the Swiss Light Source by aligning the plane grating monochromator to its zero-order reflection, such that all photon energies passed through the beamline optics and focussed onto the sample surface. The spectral profile of the PEARL non-monochromatic photon flux versus photon energy can be found in Supplementary Note 4. To maximise the incident photon flux, the front-end aperture and exit slit size of the beamline were maximised; this leads to a non-monochromatic photon flux of $3 \times 10^{14}$ ph/s, as measured by the drain current of the refocussing mirror. The non-monochromatic irradiations were repeated cyclically in 10 min intervals, with XPS measurements made after every 10 min irradiation, until a total irradiation time of 140 min was reached. Initial measurements were performed at room temperature, with no active heating or cooling being applied to the sample. The measurements were then repeated on another sample, with active cooling reaching a base temperature of 77 K. These measurements at different temperatures were carried out to check for any temperature dependent effects of the hydrogen desorption. The analysis chamber had a base pressure < $5 \times 10^{-10}$ mbar.

### Defining the photon irradiance
We define the photon irradiance as the incident photon flux per unit area, integrated over the irradiation time. This measures the total number of photons incident per unit area of the sample surface and is used to quantify the desorption yield and cross-section of hydrogen desorption. Mathematically, the photon irradiance, $\Sigma_{ph}$, is defined as:

$$\Sigma_{ph} = \frac{\Phi_{ph}}{A} t \tag{7}$$

where $\Phi_{ph}$ is the incident photon flux (ph s$^{-1}$), $A$ is the beam spot size on the sample surface (cm$^2$) and $t$ is the total irradiation time (s). Thus, photon irradiance has units of photon cm$^{-2}$ and is used throughout this work.

### X-ray filter specifications
We designed a series of three X-ray filters in collaboration with and manufactured by Luxel Corporation, Friday Harbour, WA USA. Al, Zr and C filters were chosen as they have three transmission bands in the EUV range: 17 – 72 eV, 70 – 200 eV and 100 – 280 eV, respectively[68]. The final reported film thicknesses were: (i) 256.6 nm for Al; (ii) 247.8 nm for Zr; (iii) 243.2 nm for C. The C filter consists of 137.8 nm C film with an additional 105.4 nm of LUXFilm© polyimide; the transmission behaviour remains essentially unperturbed, the only difference is that the filter is not as brittle as pure C, which makes it less likely to break during transportation. The three X-ray filters were installed onto a single stainless-steel support disk containing three slots. The outer diameter of each aperture was 5 mm, which was large enough to ensure that the incident photon beam fully passes through the filter

with little to no clipping. This stainless-steel support disk was then mounted onto a linear translator that could intercept the incident non-monochromatic beam at PEARL. Once the filters were installed and pumped down to UHV, an external 120 °C bakeout was made to desorb any residual moisture present on the surface of the filters. A plot of the calculated (provided by Luxel Corporation) and measured (at PEARL) transmittance for each filter is shown in Fig. 3c.

## Monochromatic photon irradiations

The monochromatic irradiations at PEARL (Fig. 3d) were performed in the same way as the non-monochromatic irradiations, however, the monochromatic photon flux $\approx 10^{12}$ ph/s was measured via the drain current of the refocussing mirror. The spectral profile of the PEARL monochromatic photon flux versus photon energy can be found in ref. 69. Since the SIM beamline utilises two APPLE II type undulators, an extremely high monochromatic ($E/\Delta E > 5000$) flux of $\approx 10^{14}$ ph/s is achieved[70]. During the monochromatic irradiations at SIM (Fig. 4), the exit slit of the X-ray optics was narrowed so that only a 25 μm vertical, rectangular band was irradiated. After 157 min, the exit slit was widened to span the entire field-of-view and the contrast difference between the unexposed and irradiated surface could be clearly observed with PEEM. All PEEM images were corrected by normalising the images to a corresponding flat-field image; this is a high-statistic defocused PEEM image taken over a region on the surface with no features and is used to cancel the effects of image artefacts/distortions caused by variations in the pixel-to-pixel sensitivity of the micro-channel plate detector. The start voltage acceptance window for PEEM is $\approx 0.5$ V.

## Data availability

The data that support the findings of this study are openly available on Zenodo (zenodo.org) at https://doi.org/10.5281/zenodo.7813472.

## Code availability

The codes used for plotting and fitting the data within this paper are openly available on Zenodo (zenodo.org) at https://doi.org/10.5281/zenodo.7813472.

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

## Acknowledgements

We acknowledge the excellent technical support from Patrick Ascher at the PEARL beamline, SLS, PSI. The project was financially supported by the Engineering and Physical Sciences Research Council (EPSRC) project EP/M009564/1, the EPSRC Centre for Doctoral Training in Advanced Characterisation of Materials (EP/L015277/1), the Paul Scherrer Institute (PSI) and the European Union Horizon 2020 Research and Innovation Programme, within the Hidden, Entangled and

Resonating Order (HERO) project (810451). Procopios Constantinou was partially supported by Microsoft Corporation. Part of this work was performed at the Surface/Interface: Microscopy (SIM) beamline of the Swiss Light Source (SLS), PSI, Villigen, Switzerland. We would like to thank Luxel Corporation, Friday Harbour, WA, USA for manufacturing the X-ray filters.

## Author contributions

P.C. led the experiments, data analysis and interpretation, and wrote the manuscript. S.R.S., N.J.C. and G.A. conceived and initiated the project. P.C., S.R.S., T.J.Z.S., M.M., and C.A.F.V. performed the experiments. L.T., D.K., and Y.E. discussed the results and interpretation, and revised the manuscript. All authors contributed to data analysis, interpretation, and manuscript writing.

## Competing interests

The authors declare no competing interests.
