## [Peer Review File · Nature Communications]

Editorial Note: Parts of this peer review file have been redacted as indicated to maintain the confidentiality of other journals.

REVIEWER COMMENTS

Reviewer #1 (Remarks to the Author):

I have previously declined this manuscript for publication in [redacted] since the presented results were still far from a proof of principle functional device.

After reading the revised manuscript, I commend the authors for fine tuning the text, tone down the claims and adding several important sentences. Therefore, I see this manuscript fitting well with Nature Communications and I would recommend it for publication pending addressing the following points:

1)

Pg. 2: "...developing a photon-based method, which has been the workhorse of high-volume semiconductor manufacturing, that is compatible with atomic-scale STM-based lithography has obvious technological and economic advantages".

In the end, if I understand correctly, the authors propose a laboratory where these two techniques are integrated. Would this arrangement still be industry compatible and therefore economically viable? Or it would be an incremental improvement compared to the current fabrication flow?

2)

pg 4.

"Thus, this method of photon-based hydrogen desorption lithography can potentially create metallic, large-scale contacts or interconnects to multi-layer quantum devices [14], [41]."

What really counts, in the end is the amount of phosphorus that is electrically active.

I would not forget this important detail and add the following (or equivalent wording) to the sentence "...multi-layer quantum devices, assuming that all dopants are electrically active, which may depend upon the subsequent Si encapsulation conditions."

3)

I really appreciate the new section at Pg.13 "Towards large-scale EUV patterning of silicon quantum devices"

However, I strongly recommend adding an extra figure (Fig. 5) illustrating the section. As it reads now, the readers will have a hard time imagining the proposed lab/fab flow.

Reviewer #2 (Remarks to the Author):

The manuscript describes a method to desorb H from a Si:H terminated surface. The data and analysis presented demonstrates that EUV will desorb some of the H termination. The authors perform STM image analysis coupled with extensive XPS analysis to quantify the remaining hydrogen termination after various illumination exposures. They use different illumination schemes including filters to bandwidth limit the wavelengths of importance to understand the desorption processes.

However, the interpretation of the results and potential applications of the methods fall short and call into question the implications of the results. I will discuss the shortcomings first and then go through some specific issues in the manuscript. Although this manuscript could be modified and made publishable in an appropriate journal, the questionable interpretations, extrapolations, and potential application space do not rise to the level of Nature Communications.

The premise of the manuscript is that "This is an important step towards the EUV patterning of silicon surfaces without traditional resists, by offering the possibility for parallel processing in the fabrication of silicon quantum electronic devices through deterministic dopant patterning". The manuscript does not make the claim that there are obvious applications for conventional semiconductor fabrication and manufacturing, although the authors speculate on applications for quantum information processing. The title implies the methods have application to silicon quantum electronics, based-on work by the Simmon's group and others fabricating quantum devices at the near atomic-scale using H-depassivation lithography. However, there are no demonstrations of

functional devices or contacts, but rather the manuscript focusses only on hydrogen desorption. P doping with incorporation and validation measurements would substantially improve the impact of the methods described.

It is a major technical and economic challenge to combine high resolution EUV patterning with STM hydrogen patterning as this will require the entire system, patterning, incorporation and overgrowth processes to be housed in an ultrahigh vacuum facility. Such a system does not exist today and will require enormous investment. Without a reasonable pathway to realize this kind of combined operation, the methods are limited to patterning larger scale quantum devices which do not take advantage of the key atomic-resolution aspects of STM patterning. No reasonable or practical pathway to realize combined EUV and STM limits the impact and is a major shortcoming for this manuscript.

There is great additional costs to using EUV patterning methods for contacts when ebeam patterning is successfully used by groups around the world. The main challenge for significant impact is to combine in plane patterning with STM patterning in an atomically clean environment.

In the "Towards large-scale EUV patterning of silicon quantum devices" section the authors propose to fabricate devices such as an array of quantum dots or a series of control gates for an array of atom-scale qubits. If the authors demonstrate an array of operational quantum dots, that would be a significant and more impactful manuscript. A demonstration of an operational contact or patterned metallic region would also be an important step in demonstrating the utility of the methods.

The technique as presented is limited to ~ 60 - 65% hydrogen desorption due to its electron-based activation, which immediately limits this method only to potentially making contacts to other STM patterned devices and not fabricating atomic-scale quantum devices. The authors state "The most important aspect of Figure 2b is that the values of the clean silicon atom density appear to vary non-linearly and approach 1 ML at an ever-decreasing rate." And "The clean silicon atom density gradually approaches 1 ML and to achieve a clean silicon atom density of 0.95 ML". However, the experimental data in Figure 2 only go to about irradiance of 2.75×10^{20} ph/cm² but are extrapolated to nearly 4.5×10^{20} ph/cm² in Figure 4e to make the claim of near 1 ML of clean silicon. The experimental data only demonstrate 65% silicon exposure but are inappropriately extrapolated to 95 %.

The actual hydrogen desorption is random based on the STM data and manuscript claims, and therefore the actual pathways for P incorporation and resulting P densities remain to be demonstrated. There are also concerns here that the line edge roughness and small feature

definition could be compromised due to significant line edge roughness in the resulting patterned features.

A few additional specific manuscript comments:

1) On page 4 the authors state “We quantify the surface density of clean silicon in terms of monolayers (MLs; where $1 \text{ ML} = 6.78 \times 10^{14} \text{ atoms cm}^{-2}$) by using an image threshold selection method [37] and find it to be $0.55 \pm 0.10 \text{ ML}$ for the STM data shown in Figure 1g.” As stated above, this is a very significant issue manifest in the extrapolation to 0.14 ML of P dopants. There is a significant stochastic element to dopant incorporation and effective pathways to incorporation for small patterned areas have been shown to have even lower incorporation percentages. This experimentally demonstrated limit to depassivation presents basic questions about achievable dopant densities and the ultimate feature resolution.

2) P. 4 states “this method of photon-based hydrogen desorption lithography can potentially create metallic, large-scale contacts or interconnects to multi-layer quantum devices”. While the statement is correct with the qualifier “potentially”, this has not been demonstrated and no devices were contacted. There is a very big leap from showing partial H depassivation to contacting quantum devices with conducting features.

3) On p. 8 “patterning, where multiple coherent beams are made to interfere constructively and destructively to facilitate a large-scale, high-resolution patterning of the resist” None of these methods are demonstrated or shown to be compatible with this exposure technique.

4) P. 10 again the authors state “93 eV and 106 eV irradiations performed in our XPEEM experiment at the SIM beamline closely match the best fit of the hydrogen desorption curve measured in our monochromatic and non-monochromatic exposures at the PEARL beamline (Figure 4e), where the estimated total dangling bond density also saturates at similar values.” The authors again make the important experimental demonstration of the limitations of using this method.

5) P. 12 “principal desorption mechanism is associated with electron interactions, we can make some arguments on the spatial resolution of the hydrogen desorption. The ultimate resolution is set by the interaction range of electrons with the hydrogen resist” and “Therefore, we expect that a spatial resolution of a few nanometers can be realistically achieved with EUV-based hydrogen desorption lithography.” See comments above regarding resolution due to remaining hydrogen termination atoms.

6) P. 13 “Our study demonstrates the desorption of hydrogen from the Si(001):H surface using EUV light, providing a new, resistless EUV lithography method for nanoscale device patterning and dopant localization that is compatible with STM-based fabrication and commercial device patterning.” The method as presented was not demonstrated to be compatible with commercial device patterning or any other standard commercial EUV patterning techniques.

7) P. 13 The following conclusions are not supported by the data in the manuscript. “XPEEM confirms the spatial patterning capability of this method.” And “By combining with EUV interference-based patterning techniques, this approach offers a novel way to create nanoscale, spatially confined dopant patterns in silicon for use in nanoscale devices or as interconnects for atomically precise STM patterning. The implications for the semiconductor industry are significant, as EUV lithography using 92 eV (or 13.5 nm) is now widely used...” This is speculation not supported by any experimental results.

Response to Reviewer #1

Reviewer 1 comment 1: *I have previously declined this manuscript for publication in [redacted] since the presented results were still far from a proof of principle functional device. After reading the revised manuscript, I commend the authors for fine tuning the text, tone down the claims and adding several important sentences. Therefore, I see this manuscript fitting well with Nature Communications and I would recommend it for publication pending addressing the following points.*

Our response 1: We thank the reviewer for the constructive comments and for agreeing that our manuscript now aligns well with the standards of *Nature Communications*. We appreciate the reviewers' recognition of the improvements we have made in the revised manuscript and this encouraging feedback serves as a strong motivation for us to further improve the latest version of our work.

Reviewer 1 comment 2: *1) Pg. 2: "...developing a photon-based method, which has been the workhorse of high-volume semiconductor manufacturing, that is compatible with atomic-scale STM-based lithography has obvious technological and economic advantages". In the end, if I understand correctly, the authors propose a laboratory where these two techniques are integrated. Would this arrangement still be industry compatible and therefore economically viable? Or it would be an incremental improvement compared to the current fabrication flow?*

Our response 2: The reviewer's understanding is accurate. We are proposing a laboratory setup where both EUV- and STM-based lithography are integrated. This integration aims to push the boundaries of what's currently achievable in the manufacturing of silicon quantum devices. By integrating these techniques in a controlled laboratory environment, we aim to further develop, demonstrate, and optimize the fabrication of nano-scale silicon devices using EUV lithography. As these processes become more streamlined and mature in this new setting, transitioning them to an industrial setting could become increasingly feasible. In essence, our work, as suggested by the title, is a step towards large-scale silicon quantum device patterning. This could potentially yield substantial long-term benefits for the semiconductor manufacturing industry - for instance, EUV lithography at 13.5 nm (~92 eV) is used by major players like Samsung, Intel and TSMC to target 5 nm nodes and beyond. Therefore, our work on silicon will likely be of high interest to these key stakeholders in the semiconductor industry.

While this arrangement might initially seem complex and potentially costlier than current fabrication flows, the EUV-IL system at the XIL beamline of the Swiss Light Source (SLS) already provides much of the technical equipment required, though we do acknowledge in our manuscript that there are challenges to overcome. In essence, our proposed laboratory setup is designed to circumvent some difficulties of atomic-scale manufacturing with STM. For example, to date the STM has produced (3x3) [Wang, X., Khatami, E., Fei, F. *et al.* Nat Com. 13, 6824 (2022)] and (1x10) [Kiczynski, M., Gorman, S.K., Geng, H. *et al.* Nature 606, 694–699 (2022)] 2D artificial lattices. Due to the serial patterning nature of the STM, we believe that EUV lithography can present a viable alternative for achieving much larger artificial 2D lattices. Additionally, the periodic, mutually perpendicular (criss-cross) pattern of control gates proposed for a surface code quantum computer architecture in silicon [C. D. Hill *et al.*, Sci. Adv., 1 (9) (2015)] can also be feasibly patterned with EUV lithography. Of course, we acknowledge that the atomic-scale resolution of the STM will not be matched by EUV lithography, but this is not required for all devices. The economic viability would therefore be seen in the long-term benefits of reliable quantum device manufacture at scale on silicon.

To clarify this, we have reworded the following paragraph from:

"To further explore the compatibility and resolution of EUV-based hydrogen desorption lithography, we propose comparable experiments at the EUV-IL system at the XIL beamline of the Swiss Light Source (SLS), which provides a spatially coherent beam for photon energies of up to 100 eV and is routinely used for sub-10 nm patterning [18], [19]."

To this:

"To establish the compatibility and resolution of EUV-based hydrogen desorption lithography, we propose a laboratory setup where both EUV- and STM-based lithography are integrated. This integration aims to push the boundaries of what is currently achievable in the manufacture of silicon quantum devices. While this arrangement might initially seem complex and costlier than current fabrication tools, the EUV-IL system at the XIL beamline of the SLS already meets many of the required specifications, most importantly, an intense

and spatially coherent beam of photons with energies of up to 500 eV, already in routine use for sub-10 nm patterning [18], [19].”

Reviewer 1 comment 3: 2) pg 4. “Thus, this method of photon-based hydrogen desorption lithography can potentially create metallic, large-scale contacts or interconnects to multi-layer quantum devices [14], [41].” What really counts, in the end is the amount of phosphorus that is electrically active. I would not forget this important detail and add the following (or equivalent wording) to the sentence “...multi-layer quantum devices, assuming that all dopants are electrically active, which may depend upon the subsequent Si encapsulation conditions.”

Our response 3: We appreciate the reviewer’s insightful addition here. Indeed, one of the performance metrics is the final count of electronically active dopants. The stages of PH₃ dosing, incorporation, and silicon encapsulation all significantly influence the likelihood of phosphorous becoming electrically active [K. E. J. Goh, *Physica Status Solidi (A) Applications and Materials Science*, 202(6), 1002–1005 (2005), L. Oberbeck et al., *Applied Physics Letters*, 81(17), 3197–3199 (2002), S. R. McKibbin et al. *Applied Physics Letters*, 95(23), 8–11 (2009)]. The formation of high density, electrically active phosphorus (and arsenic) layers in silicon using the hydrogen-lithography technique is already well established by many publications, including our own [T. J. Z. Stock et al. *ACS Nano*, 14(3), 3316–3327 (2020)] and many others from groups such as the Simmons group at UNSW [T. Hallam, et al. *Applied Physics Letters*, 86(14), 1–3 (2005), S. R. Schofield, *Physical Review Letters*, 91(13), 2–5 (2003)]. The only stage of the fabrication that differs in our present work is the means of hydrogen desorption. We have acknowledged in our manuscript that incomplete hydrogen desorption may lead to a reduced density of incorporated dopants and other work using e-beam lithography has shown that large doping densities are achievable [S. P. Cooil et al., *ACS Nano*, 11(2), 1683–1688 (2017)]. We do not anticipate any additional concerns regarding the electrical activity of the dopants.

We take the reviewer’s comment on board, and we have added to the following sentence in our manuscript:

“Thus, this method of photon-based hydrogen desorption lithography can potentially create metallic, large-scale contacts or interconnects to multi-layer quantum devices [14], [41]. We expect the incorporation and activation of dopants to be similar to what has been previously demonstrated on both clean surfaces [42], [43] and hydrogen-terminated surfaces patterned with STM [44], [45]. The only anticipated difference lies in the density of incorporated dopants, which we predict to be slightly lower due to any incomplete desorption of hydrogen.”

Reviewer 1 comment 4: 3) I really appreciate the new section at Pg.13 "Towards large-scale EUV patterning of silicon quantum devices". However, I strongly recommend adding an extra figure (Fig. 5) illustrating the section. As it reads now, the readers will have a hard time imagining the proposed lab/fab flow.

Our response 4: We thank the reviewer for their comment. We agree that adding an illustrative figure (Fig. 5) would greatly enhance the reader's understanding of the "Towards large-scale EUV patterning of silicon quantum devices" section on Page 13. We have now added a new Figure 5 to the manuscript, which provides a clearer picture of the proposed lab/fab flow that we have in mind. The new figure is below:

Figure 1: Illustration of a proposed laboratory process flow of patterning silicon quantum devices, divided into four stages. In the upper-right panel, we show the proposed UHV setup that can be integrated at the EUV-IL system at the XIL beamline of the SLS. The load-lock (LL) is where the silicon substrates are introduced into vacuum and then transferred to the transfer chamber. Three chambers are branched from the transfer chamber, all of which have a UHV environment with a base pressure $<5 \times 10^{-10}$ mbar: the preparation, EUV-lithography and STM chamber. The preparation chamber is used for annealing, hydrogen passivation, XH₃ dosing and silicon MBE growth. The EUV-Lithography chamber is used for the EUV hydrogen-desorption lithography. The STM chamber is used to gauge the quality of the surface at each stage and to perform atomic-scale hydrogen-desorption lithography. There are 4 main stages when it comes to fabricating silicon quantum devices; Stage 1: The process begins with preparing a clean silicon surface, followed by the atomic hydrogen passivation of Si(001). Stage 2: EUV hydrogen-desorption lithography of large-scale contacts. Stage 3: STM or EUV hydrogen-desorption lithography for patterning the nm-scale quantum device at the centre of the fiduciary markers. Stage 4: Dopant incorporation and silicon encapsulation. After this, the sample can be extracted from UHV into ambient and standard cleanroom processing techniques can be used to contact the buried device with vertical electrical connections [65].

To enhance consistency between the text and figure, we added the following text to the "Towards large-scale EUV patterning of silicon quantum devices" section on Page 13:

"With these upgrades, we foresee a laboratory with the ability to fabricate quantum devices in silicon using high-resolution EUV-based hydrogen desorption lithography. Our proposed laboratory process flow of patterning silicon quantum devices with EUV is encapsulated in Figure 5.

Figure 5 shows the basic four stages and process for patterning silicon quantum devices (stages can be repeated for fabrication of structures in three-dimensions following [62]), and we foresee that two of these stages can be effectively implemented by EUV lithography. Our proposition extends beyond the use of STM for the patterning of quantum devices and encompasses the potential application of emerging EUV lithography techniques. For example, the patterning of nanodot-arrays using EUV achromatic Talbot lithography has achieved dot sizes of 20 nm [63], [64] and we expect that it can be similarly used to fabricate devices that leverage the periodic arrangement of quantum dots to attain topological states in silicon [65]. Additionally, the periodic, mutually perpendicular (criss-cross) pattern of control gates proposed for a surface code

quantum computer architecture in silicon [14] can also be feasibly patterned with EUV lithography. Thus, in contrast to the STM, we believe that EUV lithography presents a viable alternative for achieving scalability.

In instances where there is a need to align EUV-patterned hydrogen structures with STM-patterned atomic structures, we propose an *in-situ* top-down approach: start with large-scale EUV patterning to draw large contact pads, followed by atomic-scale device patterning with the STM. This approach allows for local correction of any initial EUV pattern offset with the STM. Moreover, a proven method involves the use of pre-etched e-beam fiducial structures on the initial silicon substrate that can withstand the UHV surface preparation [66], [67] (an example of such markers is shown in Figure 5). These can also serve as reference markers for establishing electrical connections to the buried and patterned nanostructures [67]. Nevertheless, the alignment of EUV-patterned hydrogen structures with buried STM-patterned atomic structures presents a significant challenge and is an ongoing area of research.”

Response to Reviewer #2

Reviewer 2 comment 1: *The manuscript describes a method to desorb H from a Si:H terminated surface. The data and analysis presented demonstrates that EUV will desorb some of the H termination. The authors perform STM image analysis coupled with extensive XPS analysis to quantify the remaining hydrogen termination after various illumination exposures. They use different illumination schemes including filters to bandwidth limit the wavelengths of importance to understand the desorption processes. However, the interpretation of the results and potential applications of the methods fall short and call into question the implications of the results. I will discuss the shortcomings first and then go through some specific issues in the manuscript. Although this manuscript could be modified and made publishable in an appropriate journal, the questionable interpretations, extrapolations, and potential application space do not rise to the level of Nature Communications.*

Our response 1: We appreciate the reviewer's comment, but we respectfully disagree with the notion that our work leads to questionable interpretations, extrapolations, and a lack of potential application space. The reviewer correctly identifies that the core of our work is to demonstrate that hydrogen desorption lithography is a viable method for desorbing hydrogen from the silicon surface. This is evidenced by our XPS and STM data alone, and it is a significant result with applications to both conventional semiconductor fabrication and, contrary to the reviewer's view, silicon quantum information processing. We believe our work will be of interest to a wide community of scientists, semiconductor processing engineers, silicon quantum technologists, and semiconductor corporations that can benefit from the knowledge that EUV hydrogen desorption lithography is a viable technique. Therefore, our work is of great general interest and merits the broad coverage and dissemination provided by publication in *Nature Communications*. We will elaborate on the individual comments in our responses below.

Reviewer 2 comment 2: *The premise of the manuscript is that "This is an important step towards the EUV patterning of silicon surfaces without traditional resists, by offering the possibility for parallel processing in the fabrication of silicon quantum electronic devices through deterministic dopant patterning". The manuscript does not make the claim that there are obvious applications for conventional semiconductor fabrication and manufacturing, although the authors speculate on applications for quantum information processing. The title implies the methods have application to silicon quantum electronics, based-on work by the Simmon's group and others fabricating quantum devices at the near atomic-scale using H-depassivation lithography. However, there are no demonstrations of functional devices or contacts, but rather the manuscript focusses only on hydrogen desorption. P doping with incorporation and validation measurements would substantially improve the impact of the methods described.*

Our response 2: Indeed, we contend that our work provides clear applications in the field of conventional semiconductor fabrication and manufacturing, and this is one reason why we think our work warrants the broad coverage of *Nature Communications*. For instance, the current 13.5 nm (~92 eV) EUV standard for photolithography is utilized in EUV lithography by major industry players such as Samsung, Intel and TSMC to target 5 nm nodes. Therefore, our work on silicon will likely be of interest to these major players in the semiconductor industry. To make it clear that there are applications to conventional devices, we have edited the last sentence of the abstract to include:

"This is an important step towards the EUV patterning of silicon surfaces without traditional resists, by offering the possibility for parallel processing in the fabrication of classical and quantum devices through deterministic doping."

We acknowledge that we have not yet produced any functioning quantum devices using EUV-induced hydrogen desorption. As we stated in our previous response to this comment in our last review: the primary focus of our manuscript was to demonstrate, as a proof-of-principle, that hydrogen can be desorbed from the hydrogen-terminated Si(001) surface using EUV irradiation and to understand the desorption mechanism. Therefore, our manuscript shows that patterning a hydrogen-terminated Si(001) surface with EUV light is feasible and represents a first step towards large-scale silicon surface patterning. We agree that creating a fully functional device, contacted after incorporating the phosphorous dopants and conducting electrical measurements, would significantly enhance the impact of the methods. However, this requires a new laboratory which does not exist anywhere in the world. We discuss this in our "Towards large-scale EUV patterning of silicon quantum devices" section on Page 13. With the combined expertise of our collaborators at UCL, who specialize in STM lithography of atomic devices, and PSI, who are experts in EUV lithography, we believe we are uniquely positioned to realize this laboratory. This endeavour will push

the boundaries of what's currently achievable and we are in the process of applying for funding to realise this capability at the EUV-IL system at the XIL beamline. Our experiments represent the important scientific proof of concept needed to proceed with such laboratories at the SLS and elsewhere.

Reviewer 2 comment 3: *It is a major technical and economic challenge to combine high resolution EUV patterning with STM hydrogen patterning as this will require the entire system, patterning, incorporation and overgrowth processes to be housed in an ultrahigh vacuum facility. Such a system does not exist today and will require enormous investment. Without a reasonable pathway to realize this kind of combined operation, the methods are limited to patterning larger scale quantum devices which do not take advantage of the key atomic-resolution aspects of STM patterning. No reasonable or practical pathway to realize combined EUV and STM limits the impact and is a major shortcoming for this manuscript.*

Our response 3: We appreciate the reviewer's comment and understand the concerns raised. Indeed, combining high-resolution EUV patterning with STM hydrogen patterning in an ultrahigh vacuum facility is a significant challenge. However, the complexity of what we propose is not any more complicated than that of extensive academic and industrial cluster tools. These tools often incorporate more intricate growth techniques such as pulsed laser deposition, along with a series of surface analysis tools like STM, photoemission, and SIMS. We therefore disagree that there is no reasonable or practical pathway to realize such a system. Although we already discuss the technical requirements in the "Towards large-scale EUV patterning of silicon quantum devices" section on Page 13, we will further elaborate on our arguments below.

Currently, the EUV-IL system at the XIL beamline inside the SLS is a world-class beamline which provides a spatially coherent beam for photon energies of up to 500 eV and is routinely used for sub-10 nm patterning using interference lithography (see Refs. [18, 19, 59, 60]). The patterning of nanodot-arrays using EUV achromatic Talbot lithography has already been demonstrated, with dot sizes down to 20 nm (see Refs. [63, 64]). Furthermore, we have recently demonstrated in a published article the photon-induced oxide patterning of silicon, where a 75 nm half-pitch was realized (see Refs. [61]). All this previous literature has been achieved using the EUV-IL system at the XIL beamline, so the capability of such EUV lithography already exists today. We envisage that EUV hydrogen desorption lithography can also be used in a similar way on the hydrogen passivated Si(001) surface to create periodic nm-scale arrays of dangling bonds in 1D or 2D.

We have now added a **new Figure 5** to the manuscript (see Our Response 4 to Reviewer 1), which provides a clearer picture of the proposed lab/fab flow that we have in mind. To realise the combined EUV and STM capabilities that we discuss, the EUV-IL system at the XIL beamline needs to be upgraded with the following:

- A UHV compatible EUV-Lithography chamber.
- LT-STM (Scienta Omicron)
- Preparation Chamber with a Sample Heating Stage, Hydrogen Atom Beam Source (HABS), Silicon Sublimation Source (SUSI) and PH₃ / AsH₃ dopant precursor gases.

With these upgrades, we foresee a laboratory to fabricate both classical and quantum-electronic devices in silicon using high resolution EUV patterning, combined with STM hydrogen patterning. Thus, it is not unrealistic, unreasonable, or lacking any practical pathway. The technical requirements are set above, and they are achievable - what is needed is the required funding, which we are in the process of applying for based on the strength of the results presented in our manuscript.

We note that the responses listed below are also relevant to this comment:

- Reviewer 1, Our Response 2 & 4.

We hope this addresses the reviewer's concerns.

Reviewer 2 comment 4: *There is great additional costs to using EUV patterning methods for contacts when ebeam patterning is successfully used by groups around the world. The main challenge for significant impact is to combine in plane patterning with STM patterning in an atomically clean environment.*

Our response 4: We appreciate the reviewer's comment. We would like to note that this comment was made in our previous submission, and no additional rebuttal was provided by the reviewer to our previous argument. However, we again respectfully disagree with the reviewer's assertion that *'there are great additional costs to using EUV patterning methods when e-beam patterning is already successfully used by many groups around the world'*. While it's true that e-beam patterning is widely used, EUV patterning offers several distinct advantages, as we discussed in our manuscript:

"EUV photons (Figure 1a) have several technical advantages over electron beams: (i) Diffraction optics allows for interference-based parallel lithography [18], [19], which can achieve high precision and resolution over large areas; (ii) Since photons carry no charge, they interact less with residual contaminants in vacuum, resulting in cleaner devices and are insensitive to stray electric or magnetic fields in the fabrication chamber; (iii) Given the recent move towards EUV lithography by industry to target 5 nm nodes, developing a photon-based method, which has been the workhorse of high-volume semiconductor manufacturing, that is compatible with atomic-scale STM-based lithography has obvious technological and economic advantages."

Ultimately, the choice between high-volume EUV or e-beam patterning for industry will depend on a multitude of factors, including cost, performance, and the specific requirements of the application. It's also worth noting that EUV lithography is now the cornerstone of high-volume semiconductor manufacturing at and beyond the 7 nm node. As this point is already addressed in our manuscript, we have not made any additional changes in response to this comment from the reviewer.

We concur with the latter comment that one of the major challenges for achieving significant impact is the integration of in-plane EUV patterning with STM patterning in an atomically clean environment. This integration is crucial for maintaining the cleanliness of the dangling bond patterns, and for ensuring the successful fabrication of devices at the atomic scale. We also discuss this in Our Response 3 to Reviewer 2, where one of the technical requirements is a UHV compatible EUV-Lithography chamber. In contrast to electron beam lithography, which is known to introduce carbon contamination on the surface, photons carry no charge and consequently interact less with residual contaminants in vacuum, resulting in cleaner devices and are insusceptible to stray electric or magnetic fields in the fabrication chamber. This makes in-plane EUV patterning combined with STM patterning in an atomically clean environment a viable approach, as end-stations at the SLS routinely operate with a UHV environment with a base pressure $<5 \times 10^{-10}$ mbar.

Reviewer 2 comment 5: *In the "Towards large-scale EUV patterning of silicon quantum devices" section the authors propose to fabricate devices such as an array of quantum dots or a series of control gates for an array of atom-scale qubits. If the authors demonstrate an array of operational quantum dots, that would be a significant and more impactful manuscript. A demonstration of an operational contact or patterned metallic region would also be an important step in demonstrating the utility of the methods.*

Our response 5: We thank the reviewer for their comment, and we appreciate the suggestion to demonstrate an array of quantum dots, and we agree that it would significantly enhance the impact of our manuscript. We also acknowledge the importance of demonstrating an operational contact or patterned metallic region to further validate the utility of our methods. However, as both we and the reviewer have previously noted, there currently exists no laboratory worldwide equipped to carry out such demonstrations. Despite this, our work serves as a proof-of-principle for future interference lithography to pattern regular arrays of quantum dots. As we have highlighted in our manuscript, interference lithography is a well-established technique (see Refs. [18, 19, 59, 60]). Notably, the use of EUV achromatic Talbot lithography to pattern nanodot-arrays has already been showcased in the literature, achieving dot sizes down to 20 nm (see Refs. [63, 64]). By applying the same approaches outlined in these references, EUV hydrogen desorption lithography can be utilized on the hydrogen passivated Si(001) surface to create periodic nm-scale patterns. Additionally, we note that the stages of PH₃ (or AsH₃) dosing, incorporation, and silicon encapsulation is well-established in the literature (see Refs. [1,-3, 8-12, 38-40, 43-45]). The only stage of the fabrication that differs in our present work is the means of hydrogen desorption, but the rest of the steps to fabricate an operational contact is well understood. We are actively working towards this goal and as we state in Reviewer 2, Our Response 3, we need to build the necessary laboratory which does not currently exist in the world. As we previously mentioned in Reviewer 2, Our Response 2, the primary objective of our manuscript is to demonstrate, as a proof-of-principle, that hydrogen can

be desorbed from the hydrogen-terminated Si(001) surface using EUV irradiation and to understand the underlying mechanism of this desorption process.

Reviewer 2 comment 6: *The technique as presented is limited to ~ 60 - 65% hydrogen desorption due to its electron-based activation, which immediately limits this method only to potentially making contacts to other STM patterned devices and not fabricating atomic-scale quantum devices. The authors state “The most important aspect of Figure 2b is that the values of the clean silicon atom density appear to vary non-linearly and approach 1 ML at an ever-decreasing rate.” And “The clean silicon atom density gradually approaches 1 ML and to achieve a clean silicon atom density of 0.95 ML”. However, the experimental data in Figure 2 only go to about irradiance of 2.75×10^{20} ph/cm² but are extrapolated to nearly 4.5×10^{20} ph/cm² in Figure 4e to make the claim of near 1 ML of clean silicon. The experimental data only demonstrate 65% silicon exposure but are inappropriately extrapolated to 95 %.*

Our response 6: We appreciate the reviewers' perspective, but we respectfully disagree with the notion that our data has been “*inappropriately extrapolated to 95%*”. It is true that our experimental data shows a clean silicon atom density of 65% after an exposure of 2.75×10^{20} ph/cm². However, Figure 2 clearly shows that saturation has not been reached.

We'd like to clarify that our extrapolation is grounded in both the observed trend and a theoretical model of the hydrogen desorption. This model, which provides an excellent fit to our data, solves a first-order, non-linear differential equation of the form $y' \propto (1 - y)^2$ exactly and requires only one fit parameter. This model reveals that as desorption progresses, the probability of further desorption decreases, thereby slowing down the overall process. The non-linear nature of the desorption probability and the fact that it depends strongly on the remaining surface coverage of hydrogen suggests an electron-stimulated mechanism, where electrons emitted from photostimulated neighbours trigger the desorption of hydrogen. Consequently, the clean silicon atom density asymptotically approaches 1 ML as a function of photon irradiance (or irradiation time). The model, illustrated in Figure 2b, aligns remarkably well with our data, demonstrating that it offers a simple and physically accurate representation of the hydrogen desorption mechanism. There is no need for any additional parameters that would artificially constrain our fit to saturate at 65%, which just so happens to be the last data-point that was measured.

Reviewer 2 comment 7: *The actual hydrogen desorption is random based on the STM data and manuscript claims, and therefore the actual pathways for P incorporation and resulting P densities remain to be demonstrated. There are also concerns here that the line edge roughness and small feature definition could be compromised due to significant line edge roughness in the resulting patterned features.*

1) *On page 4 the authors state “We quantify the surface density of clean silicon in terms of monolayers (MLs; where 1 ML = 6.78×10^{14} atoms cm⁻²) by using an image threshold selection method [37] and find it to be 0.55 ± 0.10 ML for the STM data shown in Figure 1g.” As stated above, this is a very significant issue manifest in the extrapolation to 0.14 ML of P dopants. There is a significant stochastic element to dopant incorporation and effective pathways to incorporation for small patterned areas have been shown to have even lower incorporation percentages. This experimentally demonstrated limit to depassivation presents basic questions about achievable dopant densities and the ultimate feature resolution.*

Our response 7: We thank the reviewer for this comment; however, we note that this is a very similar comment made in our previous submission which has been already addressed in our latest revision. We elaborate on this again below.

Firstly, we acknowledge the inherent stochastic element in the pathways for phosphorous dopant incorporation. After the reviewer quotes an excerpt from our manuscript, the reviewer states that there ‘*is a significant issue manifest in the extrapolation to 0.14 ML of phosphorous dopants*’. In the very next sentence of our manuscript, we discuss this point, and we provide a quantitative estimate of the efficacy of dopant incorporation by analysing the spatial correlation of hydrogen de-passivated areas and comparing it with the known incorporation mechanisms for phosphine. Here is the full excerpt from our manuscript, where we have made a minor addition at the end for clarity:

“Although the stochastic dopant incorporation percentage is not yet known for this technique, exposing a 0.55 ML equivalent of clean silicon to phosphine can be expected to produce a maximum of 0.14 ML surface density of phosphorus, or roughly seven times greater than required for the metal-insulator transition (~0.02 ML) [40].

If we assume that all 3-dimer sites are available for phosphorous incorporation in areas desorbed with EUV light, then from our STM data we estimate an incorporation efficiency of $20\pm 10\%$ for phosphorous; this corresponds to a dopant density of 0.11 ML. Thus, this method of photon-based hydrogen desorption lithography can potentially create metallic, large-scale contacts or interconnects to multi-layer quantum devices [14], [41]. We expect the incorporation and activation of dopants to be similar to what has been previously demonstrated on both clean surfaces [42], [43] and hydrogen-terminated surfaces patterned with STM [44], [45]. The only anticipated difference lies in the density of incorporated phosphorus, which we predict to be slightly lower due to any incomplete desorption of hydrogen.

Thus, we estimate a 20% incorporation efficiency for phosphorous from our STM data by counting the available 3-dimer dangling bond sites. This estimate of the incorporation efficiency leads us to extrapolate a dopant density of 0.11 ML, which is much larger than the coverage required to reach the metal-insulator transition (~ 0.02 ML) [Ref. 40].

Smaller patterned areas may indeed exhibit lower incorporation percentages due to the configurations of the silicon dimers. It's also important to note that STM patterning will always offer superior spatial resolution compared to EUV patterning. However, with EUV lithography, we realistically anticipate achieving a spatial resolution in the range of 10's of nanometres using EUV-based interference lithography (see Refs. [59, 60, 63, 64]). Furthermore, as both we and the reviewer have previously acknowledged (see Reviewer 2, Our Response 5), there currently exists no laboratory worldwide equipped to carry out such demonstrations. These aspects will form part of our future work over the coming years where our proof-of-concept experiments will enable the construction of such a laboratory.

Reviewer 2 comment 8: 2) P. 4 states “this method of photon-based hydrogen desorption lithography can potentially create metallic, large-scale contacts or interconnects to multi-layer quantum devices”. While the statement is correct with the qualifier “potentially”, this has not been demonstrated and no devices were contacted. There is a very big leap from showing partial H depassivation to contacting quantum devices with conducting features.

Our response 8: We thank the reviewer for this comment; however, we note that this is a very similar comment made in our previous submission which has been already addressed in our latest revision.

We acknowledge that we have not yet produced any functioning devices using EUV-induced hydrogen desorption, but it is important to keep in mind the central aim of our manuscript: to demonstrate the feasibility of using photons to desorb hydrogen from hydrogen-terminated Si(001) surfaces, a previously unproven concept. Our research not only confirms, but also quantifies this possibility, opening a new avenue for the fabrication of donor-based quantum devices using EUV-based interference lithography.

We also challenge the reviewer's comment that ‘there is a very big leap from showing partial H depassivation to contacting quantum devices’. Whether it's utilizing EUV photons or electrons from a STM tip, the objective of desorbing hydrogen from a hydrogen-terminated Si(001) surface remains consistent. The well-established knowledge on how to establish contact with quantum devices in silicon can still be applied to devices patterned with EUV photons. The only distinction lies in the method of hydrogen desorption. Furthermore, our STM data indicates that the surface density of clean silicon exceeds 50%, and our longest irradiation time yields a surface density of clean silicon of 65%, as evidenced by XPS. We would not describe this as ‘partial H depassivation’, given that the majority of hydrogen is desorbed. The key point is that by utilizing a higher photon irradiance or extending the irradiation time, we can enhance this to 95% or even higher. This is further detailed in our manuscript, where we have now added a brief comparison of the expected exposure times using standard EUV sources in the industry towards the end:

“Overall, the clean silicon atom density gradually approaches 1 ML and to achieve a clean silicon atom density of 0.95 ML, a photon irradiance of 12×10^{20} ph/cm² is needed – equivalent to 18000 Joules/cm² or a three-minute exposure with an EUV (13.5nm) intensity of 100 W/cm². For comparison, modern EUV lithography sources have a power of 300 W at the intermediate focus, however, this power is significantly reduced to approximately 6 W at the wafer level due to the reflectivity loss of the optics [50]. Expectations for future systems predict a substantial increase of power at the wafer level, with predictions reaching up to 800 W for EUV sources and a reduced number of mirrors [51]. Therefore, a throughput of minutes per chip can be achieved. While this throughput might seem low compared to the production of classical devices, it is crucial to note that the scaling of computing power for quantum devices is exponential, unlike classical computers

which scale linearly. This means that quantum devices require significantly fewer qubits to surpass the performance of classical transistors.”

To avoid repetition in our responses, the responses listed below are also relevant to this comment:

- Reviewer 2, Our Response 2, 3 & 5.

Reviewer 2 comment 9: 3) On p. 8 “patterning, where multiple coherent beams are made to interfere constructively and destructively to facilitate a large-scale, high-resolution patterning of the resist” None of these methods are demonstrated or shown to be compatible with this exposure technique.

Our response 9: We thank the reviewer for their comment. However, we note that this is the same comment made in our previous submission, word-for-word with no additional rebuttal to our original reply. We respond to this comment again below.

It is important to highlight that our method’s compatibility with interference lithography is not just theoretical, given that EUV interference lithography is a well-established technique that is extensively documented and validated in the literature (see Refs. [18, 19, 59, 60]). This strong foundation together with the quantitative results in our manuscript gives us confidence in the reliability of utilising EUV interference lithography on Si(001). As also highlighted in our manuscript, the use of EUV achromatic Talbot lithography to pattern nanodot-arrays has been showcased in the literature, achieving dot sizes down to 20 nm (see Refs. [63, 64]). In addition, we have recently demonstrated photon-induced oxide patterning of silicon in a published article, where a 75 nm half-pitch was realized (see Ref. [61]). By applying the very same approaches outlined in these references in a UHV environment, EUV hydrogen desorption lithography can be utilized on the hydrogen passivated Si(001) surface to create periodic nm-scale patterns.

We note that the responses listed below are also relevant to this comment:

- Reviewer 2, Our Response 3, 4, 12.

Reviewer 2 comment 10: 4) P. 10 again the authors state “93 eV and 106 eV irradiations performed in our XPEEM experiment at the SIM beamline closely match the best fit of the hydrogen desorption curve measured in our monochromatic and non-monochromatic exposures at the PEARL beamline (Figure 4e), where the estimated total dangling bond density also saturates at similar values.” The authors again make the important experimental demonstration of the limitations of using this method.

Our response 10: We thank the reviewer for their comment. We note that this is the same comment made in our previous submission, word-for-word with no additional rebuttal to our original reply. We respond to this comment again below and this time we ensure to clarify our wording in the manuscript.

Upon reflection, we understand that the wording in our previous statement may have led to some confusion. When we mentioned that ‘*the estimated total dangling bond density also saturates at similar values*’, the word ‘*saturating*’ may be confusing here. While they do both tend towards the 1 ML limit, what we intended to convey was that the monochromatic and non-monochromatic exposures at the PEARL beamline align well with the XPEEM experiment at the SIM beamline, and the desorption rate is similar for given photon irradiances. The term ‘*saturate*’ was a remnant from our understanding prior to realizing that we could solve exactly the nonlinear differential equation for the dangling bond density, which was included in the previous submission to *Nature Communications*. This is further discussed in our response to Reviewer 2, Comment 6.

To clarify this, we have reworded this sentence from:

“We find that the 93 eV and 106 eV irradiations performed in our XPEEM experiment at the SIM beamline closely match the best fit of the hydrogen desorption curve measured in our monochromatic and non-monochromatic exposures at the PEARL beamline (Figure 4e), where the estimated total dangling bond density also saturates at similar values.”

To this:

“We find that the irradiations performed at 93 eV and 106 eV in our XPEEM experiment at the SIM beamline align closely with the best fit of the hydrogen desorption curve that was measured in our monochromatic and non-monochromatic exposures at the PEARL beamline (as shown in Figure 4e). This confirms that the total dangling bond density measured at the PEARL and SIM beamline are consistent.”

Reviewer 2 comment 11: 5) P. 12 “principal desorption mechanism is associated with electron interactions, we can make some arguments on the spatial resolution of the hydrogen desorption. The ultimate resolution is set by the interaction range of electrons with the hydrogen resist” and “Therefore, we expect that a spatial resolution of a few nanometers can be realistically achieved with EUV-based hydrogen desorption lithography.” See comments above regarding resolution due to remaining hydrogen termination atoms.

Our response 11: We thank the reviewer for their comment. We note that this is the same comment made in reply to our previous submission, word-for-word with no additional rebuttal to our original reply. We respond to this comment again below.

We concur with the referee that STM patterning offers higher resolution than EUV patterning; this will always be the case. However, as we have discussed in our manuscript and in several previous responses above, EUV patterning provides a significant advantage in terms of scalability, albeit with a trade-off in resolution. As such, STM and EUV patterning are complementary techniques, and their combination greatly expands the possibilities for fabricating large-scale quantum and classical donor devices in silicon.

We note that the responses listed below are also relevant to this comment:

- Reviewer 2, Our Response 2, 3, 5, 8 & 9.

Reviewer 2 comment 12: 6) P. 13 “Our study demonstrates the desorption of hydrogen from the Si(001):H surface using EUV light, providing a new, resistless EUV lithography method for nanoscale device patterning and dopant localization that is compatible with STM-based fabrication and commercial device patterning.” The method as presented was not demonstrated to be compatible with commercial device patterning or any other standard commercial EUV patterning techniques.

Our response 12: We appreciate the reviewer’s comment. Our manuscript demonstrates, for the first time, the desorption of hydrogen from the hydrogen-terminated Si(001) surface using EUV irradiation. The photon energies that we employ are approximately the same as the current 13.5 nm (~92 eV) which is the industry standard in EUV lithography for high volume manufacturing. While we have not explicitly demonstrated the commercial compatibility of this technique in our work, it is the methodology that will likely be of interest to the commercial device patterning industry. The key aspect of our method is its resistless nature and its ability to pattern at the nanoscale, which are desirable attributes in device lithography. To clarify this, we have reworded this sentence from:

“Our study demonstrates the desorption of hydrogen from the Si(001):H surface using EUV light, providing a new, resistless EUV lithography method for nanoscale device patterning and dopant localization that is compatible with STM-based fabrication and commercial device patterning.”

To this:

“Our study demonstrates the desorption of hydrogen from the Si(001):H surface using EUV light, thereby introducing a new, resistless EUV lithography method for nanoscale device patterning and dopant localization. This method is not only compatible with current STM-based fabrication, but also aligns with the photon energy used in commercial device patterning.”

Reviewer 2 comment 13: 7) P. 13 The following conclusions are not supported by the data in the manuscript. “XPEEM confirms the spatial patterning capability of this method.” And “By combining with EUV interference-based patterning techniques, this approach offers a novel way to create nanoscale, spatially confined dopant patterns in silicon for use in nanoscale devices or as interconnects for atomically precise STM patterning. The implications for the semiconductor industry are significant, as EUV lithography using 92 eV (or 13.5 nm) is now widely used....” This is speculation not supported by any experimental results.

Our response 13: We thank the reviewer for their comment. However, we note that this is the same comment made in our previous submission, word-for-word with no additional rebuttal to our original reply. In any case, we respond to this comment again below.

Our XPEEM data provides a demonstration that hydrogen desorption can be spatially controlled down to the 4.5 μm FWHM resolution of the optics employed, serving as a proof-of-principle for the future use of interference lithography in patterning regular arrays of fine features. Interference lithography using hydrogen-containing photoresists is a well-established technique, as we've highlighted in our manuscript (see Refs. [18, 19, 59, 60]), and additionally, the mean free path of electrons implicated in the desorption process is known to be very short. The use of EUV achromatic Talbot lithography to pattern nanodot-arrays has already been showcased in the literature, achieving dot sizes down to 20 nm (see Refs. [63, 64]). Furthermore, we've recently demonstrated photon-induced oxide patterning of silicon in a published article, where a 75 nm half-pitch was realized (see Refs. [61]). Indeed, by applying the same approaches outlined in the literature above, EUV hydrogen desorption lithography can be utilized on the hydrogen passivated Si(001) surface to create periodic nm-scale patterns. These structures can then serve as the foundation for creating a quantum device directly, such as those demonstrated by the Simmons group at UNSW (see Refs. [58]), or for patterning control gates of quantum devices that are subsequently patterned with an STM. While we do not explicitly demonstrate sub-micron EUV patterning on the hydrogen passivated silicon surface in this work, this can be inferred from our results and is supported by a wealth of evidence available in the literature.

To emphasize the XPEEM results, we added a new inset at the bottom of Figure 4c, showing that the measured FWHM of the edge is 4.5 μm and explicitly added the following sentence:

“The full width at half maximum (FWHM) across the desorption edge is measured from the XPEEM image in Figure 4c and found to be 4.5 μm .”

We note that the responses listed below are also relevant to this comment:

- Reviewer 2, Our Response 2 & 5.